# Impact of antigen test target failure and testing strategies on the transmission of SARS-CoV-2 variants

Claudia Del Vecchio[1,9], Bethan Cracknell Daniels [2,9], Giuseppina Brancaccio [1], Alessandra Rosalba Brazzale[3], Enrico Lavezzo [1], Constanze Ciavarella[2], Francesco Onelia [4], Elisa Franchin[4], Laura Manuto[1], Federico Bianca[1], Vito Cianci[5], Anna Maria Cattelan[6], Ilaria Dorigatti [2,10] ✉, Stefano Toppo [1,7,10] ✉ & Andrea Crisanti [1,4,8,10] ✉

Population testing remains central to COVID-19 control and surveillance, with countries increasingly using antigen tests rather than molecular tests. Here we describe a SARS-CoV-2 variant that escapes N antigen tests due to multiple disruptive amino-acid substitutions in the N protein. By fitting a multistrain compartmental model to genomic and epidemiological data, we show that widespread antigen testing in the Italian region of Veneto favored the undetected spread of the antigen-escape variant compared to the rest of Italy. We highlight novel limitations of widespread antigen testing in the absence of molecular testing for diagnostic or confirmatory purposes. Notably, we find that genomic surveillance systems which rely on antigen population testing to identify samples for sequencing will bias detection of escape antigen test variants. Together, these findings highlight the importance of retaining molecular testing for surveillance purposes, including in contexts where the use of antigen tests is widespread.

Despite the unprecedented approval of multiple vaccines in the two years following SARS-CoV-2 global emergence, inequitable distribution, low uptake, immune waning and immune escape by novel variants of concern (VOC) mean that population testing, which aims to identify and isolate infectious individuals to break the chain of transmission, remains central to controlling COVID-19[1]. The gold standard for SARS-CoV-2 diagnostic testing is the reverse transcription–polymerase chain reaction (RT-PCR), a molecular assay that has high sensitivity and specificity but requires laboratory analysis to amplify the genomic sequences meaning that results are often subject to reporting delays. In contrast, antigen tests are immunoassays that bind to SARS-CoV-2 proteins, typically the Nucleocapsid (N) protein, and can provide results in under 30 min. These point-of-care tests are easy to use and cost-effective, with at least 400 commercially available worldwide[2]. However, antigen test sensitivity is lower than RT-PCR, particularly when the viral load is lower (i.e., early or prior to the infectious period), resulting in more frequent false negative results[3].

[1]Department of Molecular Medicine, University of Padua, Via Gabelli, 63, Padua 35121, Italy. [2]MRC Centre for Global Infectious Disease Analysis and Jameel Institute, School of Public Health, Imperial College London, London, UK. [3]Department of Statistical Sciences, University of Padua, Via C. Battisti 241, Padua 35121, Italy. [4]Microbiology and Virology Diagnostic Unit, Padua University Hospital, Via Giustiniani 2, Padua 35128, Italy. [5]ER Unit, Emergency-Urgency Department, Padua University Hospital, Via Giustiniani 2, Padua 35128, Italy. [6]Infectious and Tropical Diseases Unit, Padua University Hospital, Via Giustiniani 2, Padua 35128, Italy. [7]CRIBI Biotech Center, University of Padua, V.le G. Colombo, 3, Padua 35131, Italy. [8]Department of Life Science, Imperial College London, South Kensington Campus, Imperial College Road, SW7 AZ London, UK. [9]These authors contributed equally: Claudia Del Vecchio, Bethan Cracknell Daniels. [10]These authors jointly supervised this work: Ilaria Dorigatti, Stefano Toppo, Andrea Crisanti. ✉e-mail: i.dorigatti@imperial.ac.uk; stefano.toppo@unipd.it; a.drcrisanti@imperial.ac.uk

On the other hand, studies have shown that antigen tests are a better correlate of culturable viruses than molecular tests[3, 4], which implies that antigen tests can more accurately identify infectious individuals[5]. There is evidence that effective screening relies on the frequency and speed of reporting more than on assay sensitivity[6]. Accordingly, the World Health Organisation recognises that despite the lower sensitivity of antigen tests, they are a cost-effective tool for a timely diagnosis that has the potential to interrupt transmission[7]. Furthermore, modelling studies comparing molecular versus antigen testing strategies across multiple transmission settings have reported that high-frequency antigen testing alone can reduce the disease burden at a lower cost than molecular testing strategies. These settings include mass testing in India[8], testing to protect in communal living settings (i.e., university halls and nursing homes)[9], and in both hospital and community resource-limited settings[10].

In October 2020, Slovakia was the first country to test its entire population using antigen tests, resulting in a notable decrease in transmission[11]. Since then, the use of antigen tests to control COVID-19 has continued to expand, despite differences in the specific policies adopted between countries[12]. Critically, in response to the Omicron VOC BA.1 in early 2022, and more recently the subvariants BA.4 and BA.5, many countries have announced testing policies that shift away from molecular testing in favour of antigen testing, including the UK[13], the USA[14], India[15], Israel[16], Canada[17], Singapore[18], Australia[18] and New Zealand[19].

The sensitivity of both antigen and molecular tests can be affected by viral mutations, which are monitored by genomic surveillance (typically by whole-genome sequencing of a proportion of RT-PCR-positive samples)[20]. Antigenic monitoring of the spike protein is routine as mutations can result in immune escape and reduced performance of molecular tests, as has been observed with both the Alpha and Omicron VOCs[21]. Whilst antigenic monitoring of the N protein has been more limited, two studies have reported variants with *N* gene mutations, which resulted in positive molecular but negative antigen test results (i.e., discordant test results)[22, 23]. So far, antigen tests have positively contributed to the surveillance of the major SARS-CoV-2 variants, as both discordant variants previously identified had limited viral fitness and circulated at low prevalence[24]. However, as antigen tests continue to play a vital role in the surveillance and control of COVID-19, there remains a risk that the absence or limited use of molecular testing may miss or select for the emergence of variants capable of escaping antigen testing. Whilst many studies have evaluated the economical and epidemiological benefits of antigen versus molecular testing strategies, none have investigated how those strategies would be impacted by a variant that can escape detection by antigen testing.

Here we first present the results of a hospital-based surveillance study in the Veneto region (Italy), during which we identified a viral variant that escapes detection by antigen tests, characterised by multiple disruptive amino-acid substitutions in the N antigen. As this variant was found to be circulating at a higher frequency in Veneto, where 57% of tests conducted between September 2020 and May 2021 were antigen, than in the rest of Italy, where 35% of tests conducted were antigen[25], we next test the hypothesis that the increased frequency of antigen testing in Veneto compared to the rest of Italy could have favoured the undetected transmission of the discordant variant.

To this end, we fit a multi-strain compartmental model of SARS-CoV-2 transmission, which accounts for population testing, vaccination, and non-pharmaceutical interventions, to the epidemiological and genomic data recorded in Veneto and in the rest of Italy, allowing us to reconstruct variant-specific incidences. We additionally test the impact of several counterfactual testing scenarios on the transmission dynamics, diagnostic test performance and genomic surveillance of escaping and concordant SARS-CoV-2 variants. Together, our results shed new light on the limitations of mass antigen testing in the absence of molecular testing, and on the importance of maintaining molecular testing, not just for diagnostic but also for monitoring and surveillance purposes.

## Results

### Analysis of antigen assay performance

Between 15 September 2020 and 16 October 2020, we conducted a hospital surveillance study at the University Hospital of Padua (Italy) to compare the performance of SARS-CoV-2 rapid antigen tests against RT-PCR molecular swabs. During the hospital study, 1441 subjects representing 44% of all patients examined (3290) in the Emergency and Infectious Diseases wards were tested with both antigen (Abbott) and molecular (Simplexa™ COVID-19 Direct Kit, Diasorin Cypress, CA, USA) tests (Fig. 1). Patients aged 0–19 were almost exclusively given the molecular test, in line with the school regulations that required a negative molecular test for school re-admission (Supplementary Figs. 1 and 2). To avoid biases, we restricted the analysis to the patients aged 20+, which accounted for 1387 subjects who took both tests and 1254 subjects who took only one test (Fig. 1).

The antigen test failed to correctly identify the presence of SARS-CoV-2 in 19 out of 61 samples that showed a clear positive signal in RT-PCR against both the S and ORF1 viral sequences. Compared to the molecular test, the Panbio™ antigen test showed an overall specificity of 99.9% (95% confidence interval (CI), 99.9–100%) and a sensitivity of 68.9% (95% CI, 55.7–80.1%) (Table 1). At Ct values below 33 for both the *S* and *ORF1* genes the sensitivity of the antigen test increased to 77.8% (95% CI, 64.4–88.0%) and 79.2% (95% CI, 65.9–89.2%), respectively (Table 1). Table 1 also presents the positive predictive value (PPV) and negative predictive value (NPV) for the antigen test at different prevalence values. Notably, we find that the PPV of the antigen test is acceptable given a prevalence of 4% (i.e., the prevalence in the analysed samples taken from the hospital's administrative databases), as well as at 0.5% prevalence; however, given a prevalence of 0.1% the PPV is <50%.

We find that the sensitivity of the antigen test increases with lower Ct values for both the *S* and *ORF1* genes (Supplementary Table 1 and Supplementary Fig. 3). The Ct value distribution of the concordant molecular +/ antigen + and discordant molecular +/ antigen – samples for both the S and ORF1 antigens are shown in Supplementary Fig. 4. Using the Welch two-sample *t*-test we found significant differences in the distributions of Ct values of the concordant and discordant samples (Supplementary Table 2). Critically, the occurrence of several molecular +/ antigen – discordant samples with Ct values well below the threshold of the antigen test sensitivity suggested the occurrence of genetic variants of the viral *N* gene that are not detected by the capture antibody reagent.

### Genotyping of concordant and discordant samples

To investigate whether the presence of genetic variants could explain the observed discordance in antigen and molecular testing in samples with high viral loads (evidenced by the low *ORF1* and *S* gene Ct values), we carried out full-length sequencing on the viral RNA present on a random subset of molecular +/ antigen – and molecular +/ antigen + swab extracts. We selected eight discordant samples (molecular +/ antigen –) and nine concordant samples (molecular +/ antigen +, as control), with S and ORF1 Ct values ranging at the time of the nasopharyngeal swab testing from 11.6 to 26.0 (see Supplementary Data 1)[26]. The observed discrepancy between the original RT-PCR values obtained on the fresh nasopharyngeal swab samples with the Diasorin kit (see Methods) and the RT-PCR values obtained from the in-house molecular method on stored swab samples at −80 °C, can be explained by the assay performance, including the different targets amplified, and the freezing-thawing process which may have affected the conservation of the samples (see Methods).

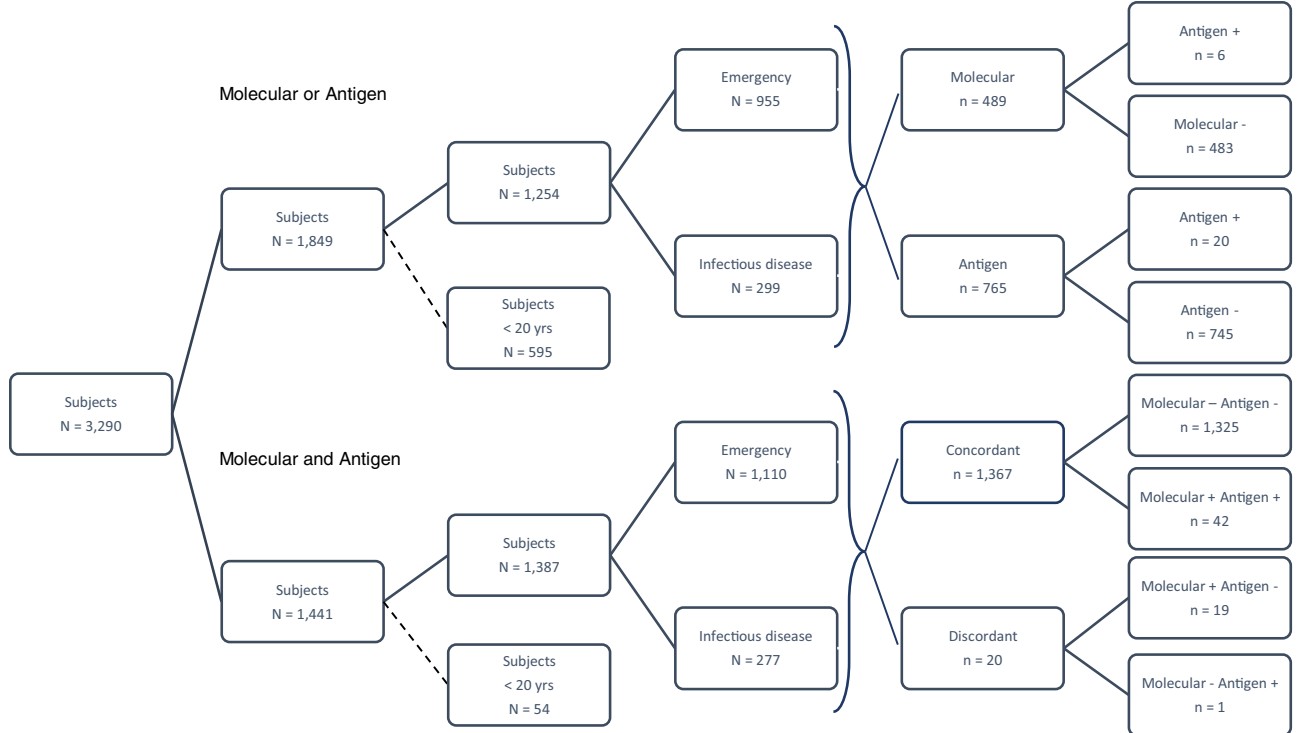

**Fig. 1 | Diagram of admissions to hospital wards.** Diagram showing the total number of patients from emergency and the infectious diseases wards tested with either a molecular or antigen diagnostic test and with both molecular and antigen tests in the period from 15 September to 16 October 2020. Subjects who underwent either antigen or molecular are further subdivided into the corresponding test group with test results. Subjects who underwent both tests are further arranged in four groups according to test result concordance (molecular –/+ antigen –/+). Subjects under 20 were excluded from the hospital surveillance study.

Analysis of assembled viral sequences confirmed that 7/8 of the discordant samples were characterised by disruptive amino-acid substitutions mapping within regions of the N protein known to contain immunodominant epitopes that function as the target of capture antibodies in antigen tests (Fig. 2)[27]. Of these, two samples were characterised by the disruptive P365S mutation (D1_8_B.1.177.7 and D2_6_B.1.177), two contained unique disruptive mutations (R209I in sample D1_7_B.1.1.119 and D348Y in sample D2_3_B.1.177.4), and three samples (D2_1_B.1.160, D2_4_B.1.160 and D2_7_B.1.160) contained viruses with the same amino-acid substitutions M234I and A376T (M234I-A376T). In addition, the eighth discordant sample D2_2_B1.1.177 contained the amino-acid substitution Q229H, which was in proximity to B-cell epitopes of the N antigen. In contrast, only 2/9 of the viruses present in the concordant samples had mutations within the region of mapped B-cell epitopes, and none showed amino-acid substitutions downstream to position 220 (Fig. 2).

From the samples sequenced, we estimate the Panbio™ COVID-19 antigen test sensitivity to be 0% (95% CI, 0–56.1%) against viruses with the M234I-A376T substitutions and 64.3% (95% CI, 38.8–83.7%) against variants not carrying the M234I-A376T substitutions (Supplementary Table 3). When considering only the viral variants without mutations in the regions of mapped B-cell epitopes, we find a test sensitivity of 87.5% (95% CI, 52.9–99.4%) (Supplementary Table 4). We further tested the ability of Abbott Panbio™ COVID-19 Ag and two additional antigen tests (COVID-19 Ag Respi-Strip from Coris BioConcept, Belgium and LumiraDX SARS-CoV-2 Antigen Test) against five cultured viruses isolated at the beginning of the pandemic in the municipality of Vo' when lineage B (original lineage first to be discovered in Wuhan province) was circulating[28]. Critically, all five isolates produced a positive signal in all antigen tests, and their sequences did not show any mutations in the N antigen (Fig. 2). Conversely, all three antigen tests failed to detect the presence of the viruses D2_3_B.1.177.4, D2_1_B.1.160 and

D1_7_B.1.1.119, despite molecular confirmation that the supernatant contained the viruses.

Supplementary Fig. 5 and Supplementary Data 1 show the frequency of both concordant and discordant virus variants in the rest of Italy (3014 analysed sequences) and Veneto region (341 analysed sequences) deposited in GISAID between 1 January 2020 and 31 December 2020[29]. We found that one conservative amino-acid substitution A220V was increasingly present in different clades in Italy. Mutations G204R and R203K appeared very early on during the pandemic, reaching 80% frequency by the end of June 2020, and declined thereafter. The amino-acid substitutions M234I-A376T, found in 38% of all discordant virus sequences we analysed, appeared in Italy (excluding Veneto) at the beginning of September 2020 and after an observed maximum prevalence of 10%, decreased to 6% by the end of December 2020 (see Supplementary Data 1). In the Veneto region, the prevalence of the amino-acid substitutions M234I-A376T peaked in December 2020 at 20%, before declining.

**Reconstructing the transmission dynamics of concordant and discordant variants**

As the use of antigen diagnostic tests began earlier, and remained higher, in Veneto compared to the rest of Italy (Fig. 3a), we next investigated the role of population testing in the transmission dynamics of a discordant variant (M234I-A376T) compared to concordant variants (A220V, the Alpha VOC and an ensemble of all other variants, Supplementary Table 5). We fit a multi-strain compartmental model (Fig. 4a) to the reconstructed variant-specific reported incidence in Veneto and the rest of Italy (Fig. 3c–f), obtained by multiplying the total reported incidence in each location (Fig. 3b) by the prevalence of the variant in GISAID, in each location (Fig. 3g–j). The antigen and molecular testing policy implemented in Italy during the modelling period (May 2020–2021), and reconstructed within the model, is presented in Fig. 4b. In line with the findings from the

**Table 1 | Performance of antigen test**

| Prevalence | All data | | | Ct S gene <33 | | | Ct ORF1 gene <33 | | |
|---|---|---|---|---|---|---|---|---|---|
| | Sample prev = 0.04 | Prev = 0.005 | Prev = 0.001 | Sample prev = 0.04 | Prev = 0.005 | Prev = 0.001 | Sample prev = 0.04 | Prev = 0.005 | Prev = 0.001 |
| Sensitivity | 68.9 (55.7–80.1) | | | 77.8 (64.4–88.0) | | | 79.2 (65.9–89.2) | | |
| Specificity | 99.9 (99.9–100.0) | | | 99.9 (99.6–100.0) | | | 99.9 (99.6–100.0) | | |
| PPV | 97.7 (98.0–99.0) | 82.1 (39.1–97.0) | 47.8 (11.3–86.7) | 97.7 (85.5–99.7) | 83.8 (42.1–97.4) | 50.8 (12.6–88.0) | 97.7 (85.4–99.7) | 84.1 (42.6–97.4) | 51.3 (12.9–88.2) |
| NPV | 98.6 (85.5–99.7) | 99.8 (99.8–99.9) | 100.0 (100–100) | 99.1 (98.5–99.5) | 99.9 (99.8–99.9) | 100.0 (100–100) | 99.2 (98.6–99.5) | 99.9 (99.8–100) | 100.0 (100–100) |

Overall sensitivity, specificity, positive predictive value (PPV) and negative predictive value (NPV) for the antigen test. In brackets, the exact binomial 95% confidence interval is reported. The PPV and NPV are presented assuming a prevalence of 0.04, as observed in the hospital surveillance study, as well as at the lower prevalence values of 0.005 and 0.001

hospital-based study described above, we assumed antigen sensitivity to be 0% against the discordant M234I-A376T variant and 68.9% against the A220V, Alpha and all other variants (i.e., the concordant variants).

We fit four different model variants of population testing, varying assumptions on access to testing (symptomatic vs. asymptomatic cases), as well as the probability of isolating given a false negative result (see Methods for a full description). Supplementary Table 6 shows the posterior mean, 95% credible interval (CrI) and log-likelihood of each model. There was no significant difference in the model log-likelihoods; however, the qualitatively best fitting model assumed that only symptomatic cases test and isolate (which is in accordance with the testing policy in Italy during the modelling study period), and is thus presented as the main analysis (Supplementary Fig. 6).

Figure 5a, b shows the transmission dynamics, obtained from the calibration of the multi-strain model described above to the reconstructed incidence of the SARS-CoV-2 variants reported in Veneto and the rest of Italy. Table 2 presents the posterior mean and 95% CrI of the parameter estimates for Veneto and the rest of Italy. We estimated the probability of a symptomatic infectious individual taking a diagnostic test to be 92.7% (95% CrI, 72.8–99.8%) for Veneto and Italy. In Veneto, we estimate that the probability of detecting an M234I-A376T variant case was significantly lower than the probability of detecting a concordant variant (Fig. 5c). For the rest of Italy, we estimate that the probability of detecting the M234I-A376T variant is comparable to the probability of detecting the concordant variants (Fig. 5c). This is in agreement with the observed prevalence of the M234I-A376T variant, which was higher in Veneto compared with the rest of Italy (Fig. 3g), a trend not seen with the concordant variants (Fig. 3h–j).

In a sensitivity analysis, we assessed the impact of assuming lower and higher estimates of antigen sensitivity against the concordant variants; different infectious and latency periods; imperfect vaccine efficacy against infection and the extent to which symptomatic individuals limit their exposure, independently of testing. Across all scenarios and all parameters, we found no significant difference in the posterior mean and 95% CrI (Supplementary Table 7). Notably, the mean probability of a symptomatic case taking a diagnostic test was 87–93% for all scenarios, with the exception of scenario 3, where we assumed antigen test sensitivity against the concordant variant to be 87.5% rather than 68.9%; under this assumption we estimated the probability of a symptomatic individual taking a diagnostic test to be lower (66.36%; 95%CrI, 25.32–99.44).

**Counterfactual analysis: impact of alternative testing strategies on cumulative incidence.** To investigate the impact of alternative testing strategies on the transmission dynamics of the concordant and discordant variants in Veneto, we explored several testing counterfactual scenarios (Fig. 5d). Under the baseline testing policy implemented across Italy during the modelling study period (molecular testing following antigen-positive tests, Fig. 4b), we estimated that 4.7% (95% CrI, 4.1–6.0%) of the susceptible population of Veneto were infected with the M234I-A376T variant between May 2020 to May 2021. This compares with 0.7% (95% CrI, 0.5–0.9%) of the susceptible population in the rest of Italy during the same period. Conversely, had the proportion of molecular and antigen tests conducted in Veneto been the same as in the rest of Italy, we estimate that 1.7% (95% CrI, 1.3–2.6%) of the susceptible population in Veneto would have been infected with the M234I-A376T variant. Our results suggest that following up negative antigen tests with a molecular test and isolating individuals receiving a positive antigen test result without molecular confirmation would have resulted in the lowest cumulative discordant and total incidence during the study period (Fig. 5d).

We estimate that antigen testing alone (i.e., without molecular confirmation), assuming an antigen test sensitivity of 68.9%, would

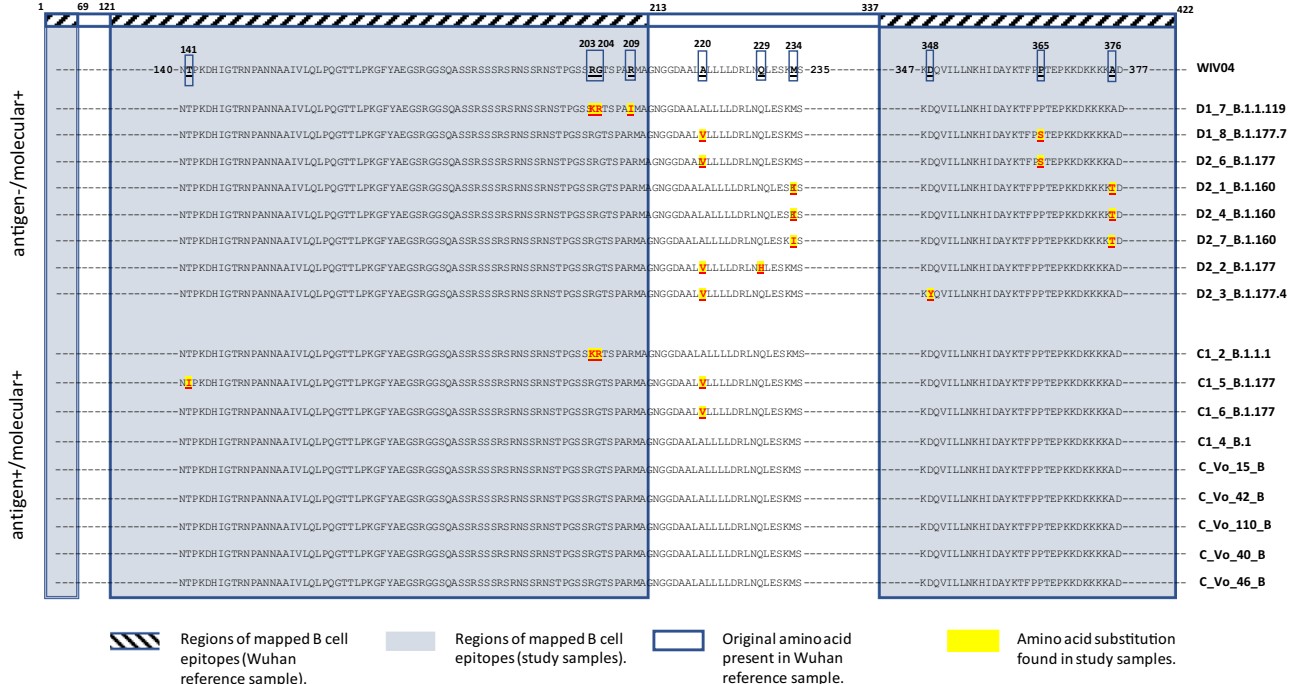

**Fig. 2 | Detected variants of N protein: the relative positions of amino-acid substitutions (with respect to the Wuhan reference sequence – WIV04) found in the N gene of 17 full-length sequences of SARS-CoV-2 are shown.** The sequences are from individuals showing either discordant, antigen –/ molecular + or concordant antigen +/ molecular +, at molecular and antigenic swab tests. Regions of mapped B-cell epitopes are highlighted on the Wuhan reference (hatched boxes) and on the sequences of the N gene (grey-shaded boxes). Only sequences around amino-acid substitutions (boxed on the Wuhan reference WIV04) are shown. Mutations found in molecular +/ antigen +, and molecular +/ antigen – sequences are shaded in yellow and highlighted in red. The last six sequences do not contain any mutation in the N gene according to the Wuhan reference sequence.

result in a cumulative incidence of 40.2% (95% CrI, 36.8–42.4%) in Veneto. This compares with a cumulative incidence of 21.1% (95% CrI, 18.2–25.5%) under the baseline testing scenario (Fig. 5d). Assuming an antigen sensitivity of 87.5% (Supplementary Tables 4 and 7) and an antigen-only testing strategy increases the transmission advantage of the discordant variant further (Fig. 5d). Figure 5e shows the cumulative total and M234I-A376T variant incidence in Veneto, under different transmissibility scenarios ($R_{0M}$) and testing strategies (molecular follows antigen + with the rates of antigen testing reported in Veneto and in the rest of Italy, as well as an antigen-only testing strategy), demonstrating that in the presence of an N antigen escaping variant, molecular testing significantly reduces the infection prevalence of the antigen escaping variant across transmission intensities.

## Diagnosis of concordant and discordant variants under alternative testing strategies
Given the impact of molecular and antigen testing strategies on the transmission dynamics of the concordant and discordant variants, we next investigated the performance of testing regimes in terms of PPV, NPV, and the probability of testing positive $p(T^+)$, given the assumed prevalence of concordant and discordant variants (Fig. 6).

In a scenario where only molecular tests are used, the PPV, NPV and $p(T^+)$ are high and independent of whether a variant is concordant or discordant (Fig. 6f). On the other hand, antigen testing without molecular confirmation has a <2% $p(T^+)$ and a 0% PPV when the concordant prevalence is <0.5%, irrespective of the discordant prevalence (Fig. 6a); there is a >25% risk of both false positives and false negatives (see Methods) when the discordant prevalence is moderate to high. The high risk of false positives observed when the concordant prevalence is low can be mitigated by confirming the antigen-positive tests with a molecular test (Fig. 6c). However, following up antigen-positive tests with a molecular test has limited impact on the NPV and $p(T^+)$, irrespective of the proportion of tests that are followed up (Fig. 6b, c). On the other hand, following up 50% of antigen-negative tests with a molecular test increases the PPV, NPV and $p(T^+)$ compared to antigen testing alone (Fig. 6d). Furthermore, following up all antigen-negative tests with a molecular test can achieve an NPV equal to or higher than that of the molecular testing scenario (Fig. 6e, f).

## Genomic detection of concordant and discordant variants under alternative testing strategies
Finally, we investigated the impact of antigen and molecular testing scenarios on the detection of variants through genomic surveillance, assuming that 0–3% of samples are selected for genomic sequencing. In testing scenarios which rely solely on antigen tests for diagnostic and surveillance purposes, we found that genomic sequencing of both antigen-negative and antigen-positive specimens is required to provide an unbiased estimate of both concordant and discordant variant incidence (Fig. 7a–c). For instance, at a prevalence of 10%, sequencing 0.5% of cases would detect 0.025% of both concordant and discordant cases. Molecular-only testing followed by sequencing of positive samples can also provide an unbiased estimate of concordant and discordant variant incidence and increases the percentage of cases detected, compared to the antigen-only scenario (e.g., 0.05% cases detected given sequencing of 0.5% of samples and 10% prevalence, Fig. 7d).

Genomic surveillance in a testing scenario where positive antigen tests are confirmed with a molecular test and positive molecular specimens are sequenced will fail to detect a discordant variant, irrespective of the proportion of specimens sequenced (Fig. 7e). Conversely, molecular confirmation of 50% of negative antigen tests followed by genomic sequencing of 0.5% of positive molecular specimens can detect 0.01% and 0.02% of concordant and discordant cases at 10% prevalence, respectively (Fig. 7f).

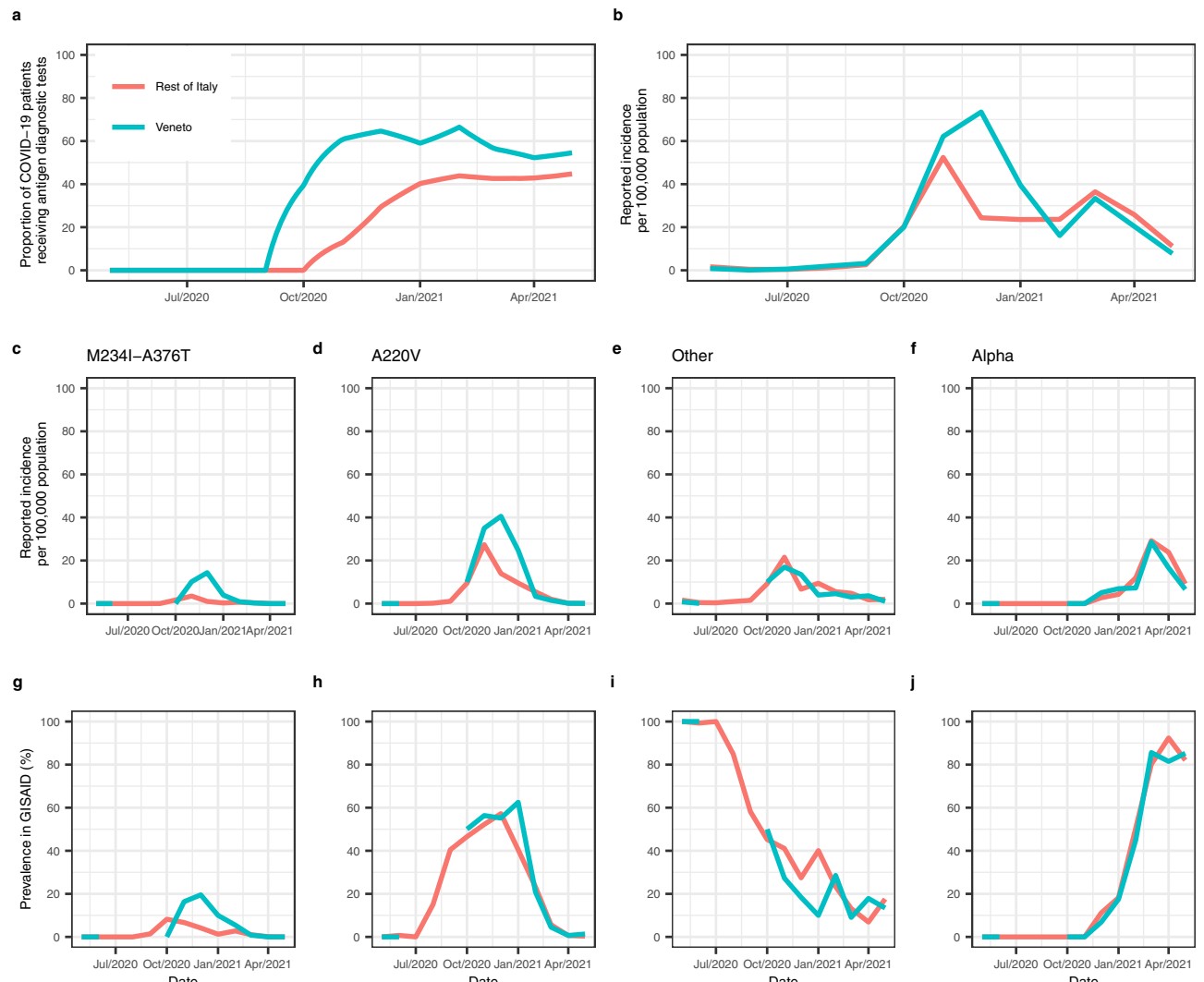

**Fig. 3 | Observed epidemiological data from Veneto and the rest of Italy during the modelling study period. a** Proportion of COVID-19 patients receiving antigen diagnostic tests, **b** total daily reported incidence per 100,000 population, **c–f** variant-specific daily reported incidences per 100,000 population, reconstructed by multiplying the mean daily reported incidence for each month by the variant-specific prevalence reported in GISAID for each month, **g–j** Variant-specific prevalence reported in GISAID, with the variant specified in panels **c–f**, respectively. GISAID Global Initiative on Sharing Avian Influenza Data.

## Discussion

This work reports a pattern of disruptive amino-acid substitutions within a major B-cell epitope of the N antigen[27] that escapes detection by the capture antibody of antigen tests. SARS-CoV-2 variants carrying these mutations were detected at a high frequency in Veneto, the first Italian region to adopt the use of antigen tests for diagnosis and surveillance. Using a data fusion approach and model calibration to the observed genomic and epidemiological data, we found that the use of antigen testing in Veneto limited detection of the discordant (molecular +/antigen −) variant and favoured its spread compared to the rest of Italy, where the uptake of antigen testing was slower and remained lower throughout the modelling study period.

Previous studies comparing antigenic and molecular testing regimes to control COVID-19 report that antigen testing alone is an acceptable alternative to strategies that include molecular testing[8–10]. In agreement, we found that antigen-only testing at a sufficient sensitivity can reduce the concordant variant incidence, compared to scenarios where molecular testing is also used. However, in the presence of variants with disruptive amino-acid substitutions within the N antigen that can escape antigen testing, this strategy can nevertheless result in an increased total infection burden. As SARS-CoV-2 incidence

remains high globally, novel variants will emerge, including VOCs[30]. This study highlights how the use of only antigen testing could facilitate the transmission of new variants escaping antigen detection, which has important public health implications in resource-limited settings, where antigen testing is a cheaper and more feasible alternate to molecular testing[31], as well as in high-income settings where molecular testing has declined rapidly[32]. Investment in molecular testing capacity, including the rapid update of RT-PCR primer sets able to detect new circulating VOCs, and strengthening of epidemiological surveillance should therefore remain a global priority, to facilitate the rapid detection and control of discordant variants.

Genomic surveillance remains critical to conduct antigenic monitoring and identify novel SARS-CoV-2 variants[33–35]. At the time the discordant variant was detected, Italy sequenced an average of 1.4% of positive samples; however, for the first 6 months of 2022 <0.3 % of cases were sequenced[36], falling short of the suggested benchmark of 0.5% of cases[37]. Here we show how molecular confirmation of positive antigen tests (i.e., the testing strategy adopted in Italy and the UK during the modelling study period) results in under-reporting of a discordant variant to genomic surveillance. This finding coupled with reduced genomic surveillance is concerning, as accurate prevalence

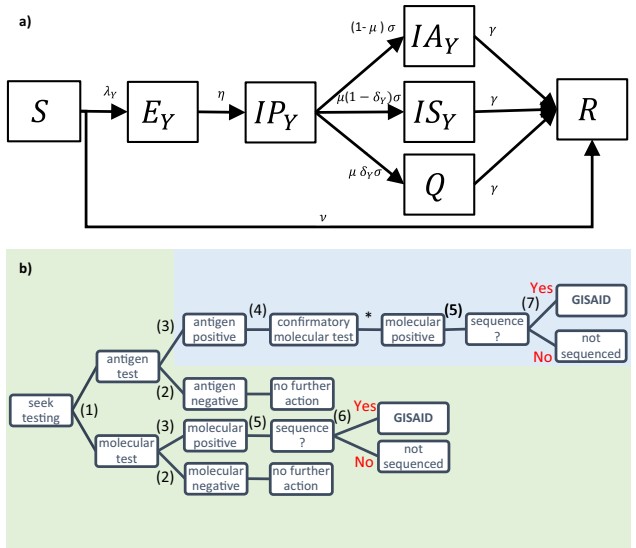

**Fig. 4 | Transmission model and reconstruction of the SARS-CoV-2 reporting process in Italy. a** Simplified flow diagram of the compartmental model used to reproduce the dynamics of the discordant variant M234I-A376T and the concordant variants A220V, Alpha, and other variants, in Veneto and the rest of Italy. Susceptible individuals (S compartment) are infected at rate $\lambda_Y = \frac{\beta_Y(IP_Y + IA_Y + IS_Y)}{S_0}$, where subscript $Y$ refers to the virus variants (M234I-A376T, A220V, Alpha, other variants). Upon infection, the latency period ($E_Y$ compartment) lasts for an average of $1/\eta$ days after which individuals are infectious but asymptomatic ($IP_Y$ compartment) for an average of $1/\sigma$ days. We assume that a proportion $(1 - \mu)$ of infections remain asymptomatic ($IA_Y$ compartment) whilst the remaining proportion ($\mu$) develop symptoms; of these symptomatic individuals we assume that a proportion $(1 - \delta_Y)$ are not detected by surveillance ($IS_Y$ compartment) and the remaining proportion $\delta_Y$ is detected, reported and isolates (Q compartment). After an average infectious period of $1/\gamma$ days individuals recover and test negative (R compartment). Susceptible individuals are vaccinated and enter the R compartment at rate $v$. **b** Description of the Italy testing policy, reproduced in the compartmental model. (1) Symptomatic individuals present for diagnosis with either an antigen or molecular test. (2) Negative test results warrant no further action and individuals contribute to transmission (IS compartment). (3) Individuals with a positive antigen or molecular test result isolate until their recovery (Q compartment, 100% compliance). (4) Positive antigen tests are additionally confirmed with a molecular test. (5) A random proportion of molecular-positive samples are selected for genomic surveillance and reported in GISAID. (6) In the green pathway, both discordant and concordant variant samples will be reported in GISAID. (7) In the blue pathway, only concordant variant samples will return a positive antigen result and be reported in GISAID. As positive antigen cases are confirmed by molecular tests, the probability of reporting a concordant variant in GISAID is independent of the test administered. For the discordant variant, the probability of reporting in GISAID depends on the probability that the initial test is molecular. *Antigen-positive samples are assumed to also be molecular-positive.

estimates of new variants are necessary to ensure appropriate interventions can be put in place[38]. Moreover, it has been shown that effective genomic surveillance depends more on the population coverage of diagnostic testing than on the proportion of positive samples sequenced, as high testing rates are required to ensure representative sampling of circulating variants[39]. To this end, increased rates of population testing, as well molecular testing independent to antigen test confirmation, for instance as the first-line diagnostic tool in hospitals[40] or through regular nationwide surveys[41], are integral for unbiased and early detection of a discordant variant.

Currently, positive samples may also be selected for sequencing based on molecular testing characteristics like spike gene target failures, which can indicate immune evasion[42]. Our work highlights the importance of also monitoring N antigen diversity, to assess how new mutations may limit the performance of antigen tests. Finally, it was recently shown that SARS-CoV-2 genome sequencing from antigen

tests is possible[43]. If this were to become a widely used surveillance tool, our study suggests that sequencing positive and negative antigen samples would be the optimal surveillance strategy to accurately detect both concordant and discordant variants.

In the event of a novel discordant variant, effective testing and tracing plays a vital role in slowing its emergence and spread. Our results indicate that the performance of antigen-only testing strategies will be poor in such a scenario. High discordant variant transmission coupled with the low concordant variant transmission is expected to produce a high proportion of both false positive and false negative results. Whilst false positive results can likely be identified through repeated antigen testing[44], this strategy is ineffective at reducing the number of false negative results in the presence of a discordant variant, which can instead be detected by molecular confirmation of negative antigen tests. The optimal proportion of tests to follow up depends on the local SARS-CoV-2 prevalence, economic and logistical considerations, and the antigen testing strategy being implemented.

Several assumptions underpin this study. Firstly, we assumed that antigen test sensitivity is 0% against the discordant variant, in agreement with our observations and another study reporting variants with 0% antigen test sensitivity[23]. However, it is possible that some antigen tests may detect a discordant variant if they use antibodies that recognise different N antigen epitopes or target different proteins all together. In this instance, the performance of antigen testing strategies against discordant variants would be >0%. We also assumed a concordant test sensitivity of 68.9% based on the hospital study, although we explore the impact of higher and lower antigen test sensitivity estimates in a sensitivity analysis. We also assumed that the prevalence of a variant in GISAID is representative of its population-level prevalence, which may not always be the case. For instance, the Alpha variant may have been oversampled due to its characteristic S-gene-target-failure, further highlighting the need for robust and unbiased genomic surveillance systems. In our main analysis, we assumed that only symptomatic individuals undertake diagnostic testing; however, in a sensitivity analysis, we explored alternative testing scenarios. Finally, we assumed homogeneous mixing within the population as the lack of age-specific surveillance data limited the extent to which we could account for heterogeneous contact patterns by age, for instance.

Rapid diagnostic testing and genomic surveillance have been a central component in the response to the COVID-19 pandemic and are likely to be vital for future pandemic preparedness as well. Our work demonstrates how the widespread use of antigen tests could facilitate the transmission of novel emerging variants capable to evade N antigen detection, thus undermining the efforts of population testing. We additionally highlight how genomic surveillance that relies on antigen tests to identify cases for genomic sequencing can bias and underestimate the prevalence of escaped antigen test variants. Retaining molecular testing for surveillance is essential in this and future phases of the pandemic, to ensure the timely detection and control of emerging SARS-CoV-2 variants globally.

## Methods
### Hospital-based surveillance
**Study design.** This study is a retrospective analysis of routinely collected data including swab samples collected from 15 September 2020 to 16 October 2020 from patients admitted to the Emergency and Infectious disease wards of the University Hospital of Padua (Italy) and from subjects who required SARS-CoV-2 testing for one of the following reasons: (a) presence of symptoms indicating a possible SARS-CoV-2 infection (fever and/or cough and/or headache, diarrhoea, asthenia, muscle pain, joint pain, loss of taste or smell, or shortness of breath, with or without pneumonia); (b) patients who were asymptomatic but had a contact with a confirmed case of SARS-CoV-2 infection during the previous ten days. All subjects >19 years were tested within

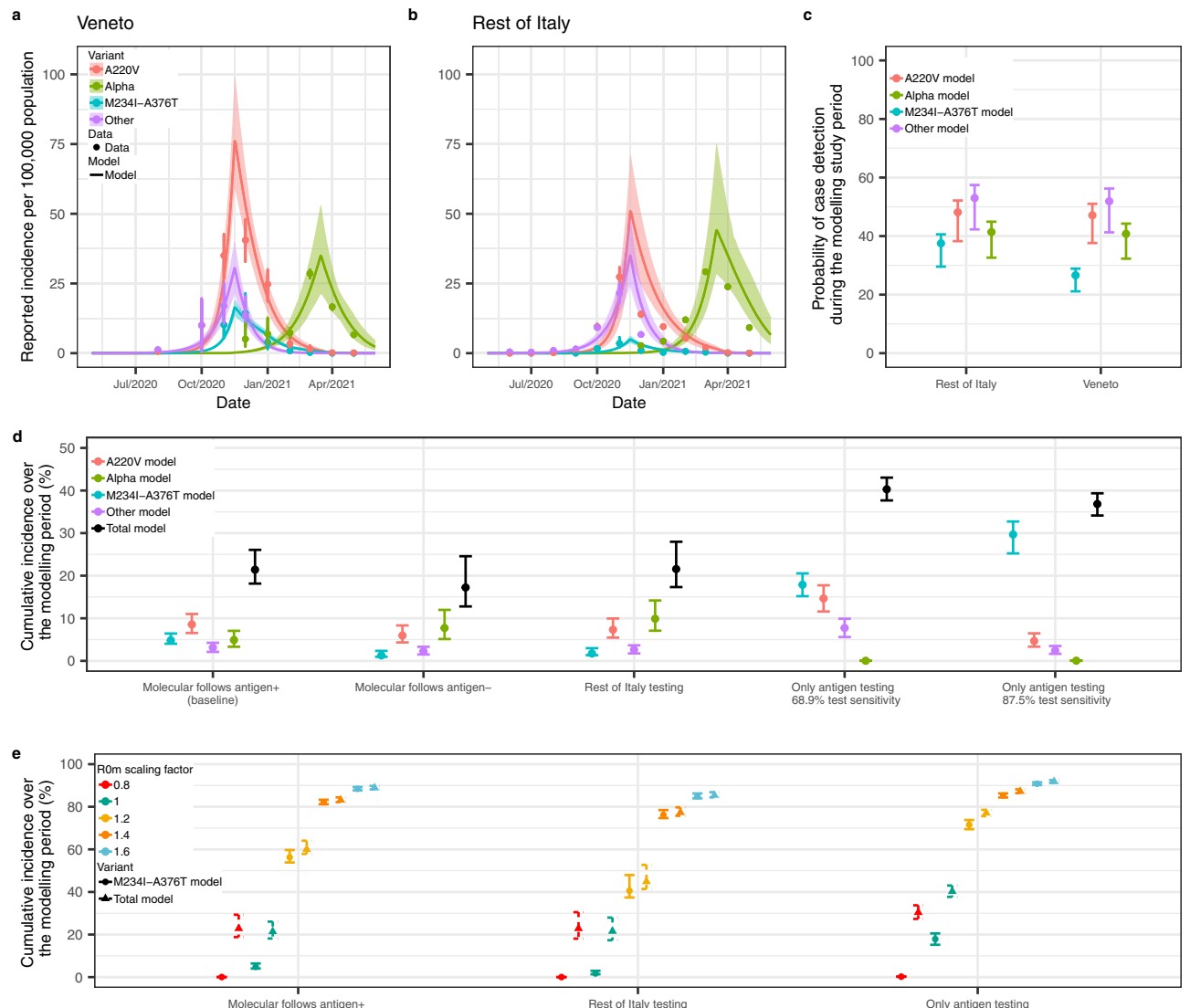

**Fig. 5 | Transmission dynamics of SARS-CoV-2 variants (A220V, M234I-A376T, Alpha and others) in Veneto and the rest of Italy, and cumulative incidences under different counterfactual testing scenarios. a** Veneto and **b** the rest of Italy estimated daily reported incidences per 100,000 population of A220V, M234I-A376T, Alpha and other virus variants, fitted from observed incidence data assuming a Negative Binomial likelihood (point and 95% binomial CI). Estimates are the mean and 95% CrI (solid line and shaded region), obtained from the 2.5% and 97.5% percentiles of 100 samples of the posterior distributions. **c** Probability of detecting cases attributable to the A220V, M234I-A376T, Alpha or other virus variants in Veneto and the rest of Italy during the modelling study period. Point and error bars are the mean and 95% CrI obtained from the 2.5% and 97.5% percentiles of 100 samples of the posterior distributions. **d** Cumulative A220V, M234I-A376T, Alpha, other virus variants and total incidence (%) in Veneto during the modelling study period, under different counterfactual testing scenarios. **e** Cumulative M234I-

A376T variant and total incidence (%) in Veneto during the modelling study period, under different counterfactual testing scenarios. The $R_0$ of the M234I-A376T variant is multiplied by different scaling factors: 0.8, 1 (baseline), 1.2, 1.4, 1.6. The counterfactual testing scenarios are as follows: molecular follows antigen + is the baseline testing scenario conducted in Veneto; molecular follows antigen – assumes negative antigen tests are followed up with a molecular test; Italy testing assumes the proportion of antigen and molecular tests conducted in Veneto was the same as the rest of Italy over the modelling study period; only antigen testing, 68.9% test sensitivity assumes infections are tested with an antigen test of 68.9% sensitivity (without molecular confirmation); only antigen testing 87.5% test sensitivity assumes infections are tested with an antigen test of 87.5% sensitivity (without molecular confirmation). Point and error bars are the mean and 95% CrI obtained from the 2.5% and 97.5% percentiles of 100 samples of the posterior distributions. CI confidence interval, CrI Credible interval.

1 h with both antigen (Panbio™ COVID-19 Ag Rapid Test Device, ABBOTT Lake Country, IL, USA) and molecular (Simplexa™ COVID-19 Direct Kit, Diasorin Cypress, CA, USA) swab assay. No data were collected on patients receiving the same test more than once. Information was collected from each individual about their age, sex, date of sampling, symptoms, time of symptom onset and Ct value of the molecular test. The molecular and antigen test results were grouped as concordant (molecular +/ antigen +) or discordant (molecular +/ antigen –). Individuals 0–19 years old were predominantly tested with molecular swabs only in compliance with local regulations and the school admittance policy (negative molecular test for re-admission to school

upon (a) or (b), as described above) in place at the time; the few that were tested with both antigen and molecular swabs were omitted from the analysis to avoid sampling biases.

**Testing for SARS-CoV-2 with antigenic and molecular assays.** Two swabs were collected from each patient. One sample was processed with Panbio™ COVID-19 Ag Rapid Test right after sampling according to the manufacturer's instructions. The second swab was processed with RT-PCR, DiaSorin Molecular Simplexa™ COVID-19 Direct assay system (Diasorin Cypress, CA, USA) that amplifies two targets of the SARS-CoV-2 genome (the *S* gene and the *ORF1ab* gene). The assay also

**Table 2 | Mean and 95% CrI parameter estimates, obtained from the 2.5% and 97.5% percentiles of the posterior distributions**

| Parameter | Mean (95% CrI) | Parameter description |
|---|---|---|
| $R_{OM}$ | 1.38 (1.3–1.43) | Transmission rate of M234I-A376T variant |
| $R_{OA}$ | 1.45 (1.39–1.51) | Transmission rate of A220V variant |
| $R_{OO}$ | 1.26 (1.23–1.31) | Transmission rate of other variants |
| $R_{OAl}$ | 2.26 (2.16–2.37) | Transmission rate of Alpha variant |
| $\rho$ | 92.65% (72.84–99.82) | Probability of performing a diagnostic test if symptomatic |
| $\omega_1$ | 36.56% (33.68–39.16) | Per cent reduction in transmission after control measures (second wave) |
| $\omega_2$ | 51.77% (47.29–56.01) | Per cent reduction in transmission after control measures (third wave) |
| $IO_M Italy$ | 105.41 (54.63–167.04) | Initial number of M234I-A376T variant infections |
| $IO_A Italy$ | 0.59 (0.06–2.42) | Initial number of A220V variant infections |
| $IO_O Italy$ | 88.27 (21.26–193.9) | Initial number of other variant infections |
| $IO_{Al} Italy$ | 88.34 (31.44–180.4) | Initial number of Alpha variant infections |
| $IO_M Veneto$ | 24.68 (13.5–37.19) | Initial number of M234I-A376T variant infections |
| $IO_A Veneto$ | 0.07 (0.01–0.27) | Initial number of A220V variant infections |
| $IO_O Veneto$ | 6.29 (1.44–14.89) | Initial number of other variant infections |
| $IO_{Al} Veneto$ | 21.86 (11.31–34.84) | Initial number of Alpha variant infections |
| $\kappa$ | 0.05 (0.01–0.12) | Overdispersion parameter |

reveals the presence of host mRNA in the same reaction to confirm the correct execution of the test. Samples showing a positive result for both viral targets were considered positive. Samples with either a single positive target or with Ct value ≥33 were confirmed with an in-house real-time RT-PCR targeting the *N2* gene[45], if this was also positive then the sample was considered positive. Sequences of oligonucleotides and probes (Company name: Thermo Fisher Scientific 168 Third Avenue Waltham, MA 02451, USA): 019-nCoV_N2 Forward Primer (TTA CAA ACA TTG GCC GCA AA), 2019-nCoV_N2 Reverse Primer (GCG CGA CAT TCC GAA GAA), 2019-nCoV_N2 Probe (FAM-ACA ATT TGC CCC CAG CGC TTC AG-BHQ1).

**Viral culture.** Viral isolation was carried out within a biosafety-level 3 containment facility. Typically, 100 µl of not inactivated swab medium were inoculated in monolayers of Vero cells (ATCC CCL-81) with Dulbecco's Modified Eagle Medium (Gibco) supplemented with 2% (v/v) of foetal bovine serum (Gibco) and 1% (v/v) of penicillin/streptomycin (Gibco) and incubated at 37 °C with 5% $CO_2$. Twenty-four hours after inoculation culture supernatant was replaced with fresh culture medium. The supernatant was collected and centrifuged and the presence of SARS-CoV-2 was confirmed by in-house real-time RT-PCR targeting the *N2* gene. Supernatant was also tested with Panbio™ COVID-19 Ag Rapid Test, COVID-19 Ag Respi-Strip (Coris BioConcept, Belgium) as well as with LumiraDX SARS-CoV-2 Antigen Test.

**Samples chosen for sequencing.** The samples stored at −80 °C and chosen for sequencing have been extracted and reamplified with the in-house real-time RT-PCR targeting the *N2* gene to check for their integrity.

**Synthesis of cDNA and library preparation protocol.** Total nucleic acids were purified from 200 µl of nasopharyngeal swab samples and eluted in a final volume of 100 µl by using a MagNA Pure 96 System (Roche Applied Sciences). Negative extraction control was also included. Then, the extracts were treated with DNase using the DNA-free™ Kit (Ambion, Life Technologies) following the manufacturer's instructions prior to cDNA synthesis. Negative controls were included

also in the following steps up to sample sequencing checking for potential contaminations. The cDNA was synthesised using the ProtoScript® II First Strand cDNA Synthesis kit (New England Biolabs Inc.). The first strand synthesis reaction was performed in a total volume of 20 µl at 95 °C for 5 min, then the temperature ramped down to 20 °C. The cDNA synthesis was completed at 25 °C for 5 min followed by 1 h at 42 °C and 5 min at 80 °C. The cDNA libraries were prepared following the Twist Library Preparation Kit for ssRNA Virus Detection protocol (Twist Bioscience) and fragmented to generate 300 bp dA-tailed DNA fragments. The Twist Universal Adapters were ligated to the dA-tailed DNA fragments to prepare the cDNA libraries ready for indexing. The samples were amplified using the Twist Unique Dual Index Primers for 10 cycles, set up as follows: denaturation step at 98 °C for 25 s; annealing step at 60 °C for 30 s and extension step at 72 °C for 30 s. The PCR products were then purified using the DNA Purification Beads (Twist Bioscience).

**Viral cDNA enrichment protocol and sequencing.** The library's enrichment was performed according to the Twist Target Enrichment protocol (Twist Bioscience). The samples were pooled in groups of four to a final concentration of 1500 ng. The hybridisation reaction to SARS-CoV-2 specific probes was performed by incubating the library pools and the probes at 70 °C for 16 h. The hybridised targets were then captured by streptavidin beads following the manufacturer's instructions. The library pools were then enriched through a post-capture PCR. The amplification products were purified using the DNA Purification beads (Twist Bioscience). After quantification and validation check, libraries were normalised and sequenced 2 × 150 paired-end on both the Illumina MiSeq and NextSeq 550 platforms at the Polo GGB facility (Siena, Italy). The sequencing was performed using the Standard V2 and the Mid Output V2.5 flowcells, respectively.

**Quality check and mapping of the reads.** Raw sequences were filtered for length and quality with Trimmomatic v0.40[46] according to the following parameters: ILLUMINACLIP: TruSeq3-PE-2:2:30:10 LEADING:30 TRAILING:30 SLIDINGWINDOW:4:20 MINLEN:90. High-quality reads were further filtered for the presence of non-specific captured sequences from bacterial, viral, and human genomes during library construction. The remaining reads were then aligned on the SARS-CoV-2 reference genome (GenBank ACC: NC_045512 [https://www.ncbi.nlm.nih.gov/nuccore/1798174254]) with BWA-MEM v0.7.17[47]. Duplicated reads were then removed with Picard tool v2.25.0 (http://broadinstitute.github.io/picard/). Consensus sequences were generated using a combination of SAMtools v1.11[48] and VarScan v2.4.1[49] variant caller. Consensus sequences were reconstructed from VarScan output with an in-house script that automatically introduces 'N' in low-quality or uncertain/uncovered regions of the reference sequence. Sequence details are available in Supplementary Data 1. Sequences were submitted to GISAID[29] and GenBank[50] (GISAID and GenBank accession numbers are available from Supplementary Data 1).

**Analysis of the amino-acid variants in N protein.** Sequence data and corresponding metadata, released from 1 January 2020 to 17 February 2021, were downloaded from GISAID databank[29] and further analysed to map the variants in N protein. The list of used sequences is reported in Supplementary Data 2. Sequences with missing information and incomplete N protein were discarded. The analysis was conducted on the variants found in concordant (molecular +/ antigen +) and discordant cases (molecular +/ antigen −) (see Supplementary Data 1). The monthly prevalence trend of the variants found in concordant and discordant cases was plotted from January to December 2020 and reported for Italy, and the Veneto region (Supplementary Fig. 5).

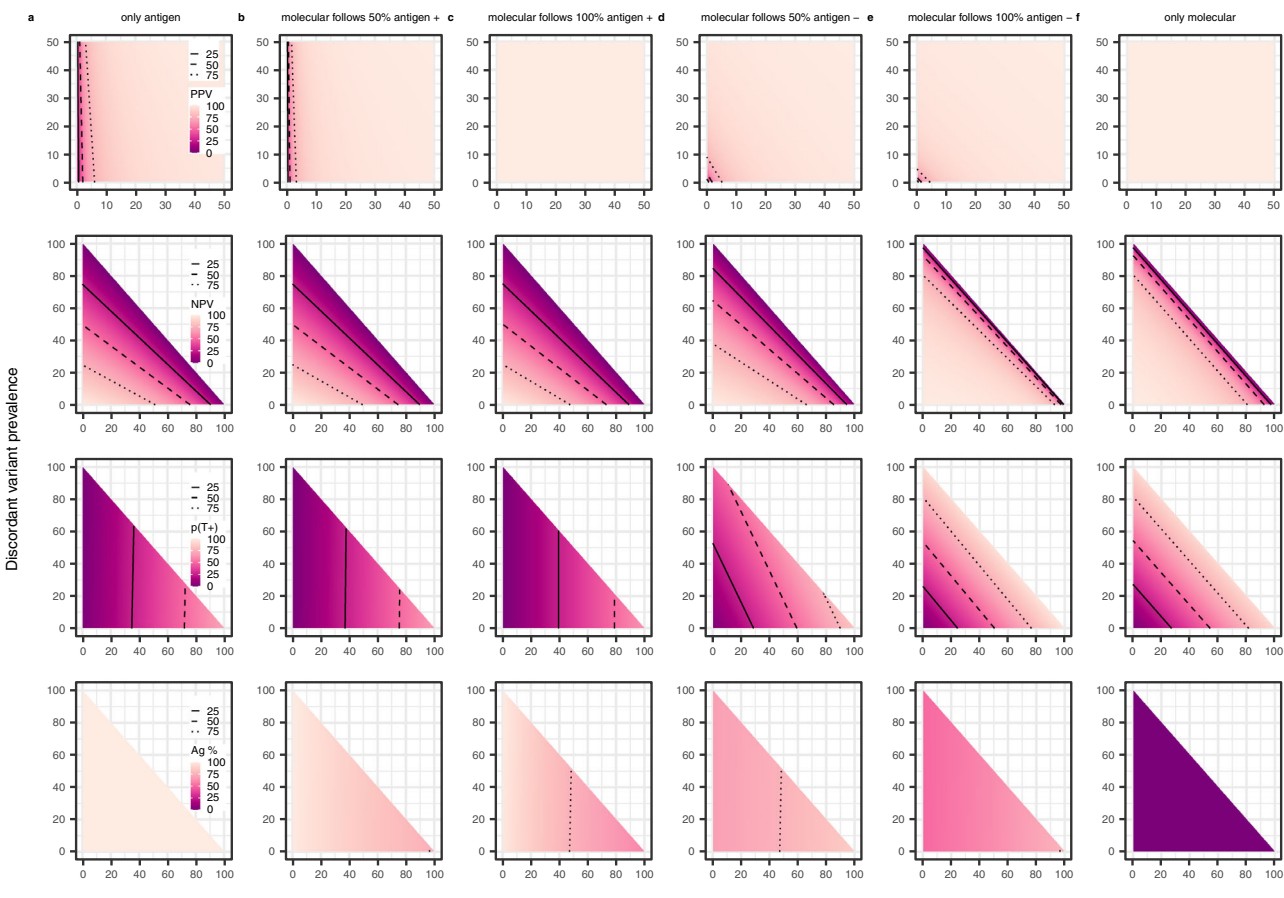

**Fig. 6 | Performance of antigen and molecular testing strategies, given a range of concordant and discordant variant prevalence values. a** Individuals present for diagnosis with an antigen test and there is no molecular test confirmation. **b** Individuals present for diagnosis with an antigen test and 50% of positive cases are confirmed with a molecular test. **c** Individuals present for diagnosis with an antigen test and all positive cases are confirmed with a molecular test. **d** Individuals present for diagnosis with an antigen test and 50% of negative cases are confirmed with a molecular test. **e** Individuals present for diagnosis with an antigen test and all negative cases are confirmed with a molecular test. **f** Individuals present for diagnosis with a molecular test. Presented are the positive predictive value (PPV), negative predictive value (NPV), probability of testing positive ($p(T^+)$) and percentage of diagnostic tests that are antigen (Ag %). Prevalence values range from 0 to 100% for both the discordant and concordant variants. Note that for PPV (top panel) only prevalence values from 0 to 50% are presented. Molecular sensitivity and specificity are 92% and 100%, respectively, regardless of the infecting variant. Antigen sensitivity is 68.9% against the concordant variant and 0% against the discordant variant. Antigen specificity is assumed fixed at 99.68% against both variants. Solid contour denotes 25%, dashed contour denotes 50% and dotted contour denotes 75%.

**Statistical analysis**. *P* values are evaluated at the 5% significance level unless otherwise specified; 95% CIs were estimated using the exact binomial method. Continuous outcomes were compared using the two-sided Wilcoxon–Mann–Whitney test (two groups) and the Kruskal–Wallis test (>two groups)[51]. Pearson's $\chi^2$ test statistic was used to compare proportions. The distribution of Ct values was estimated using a two-component Gaussian mixture. Logistic regression is used to model antigen test sensitivity against Ct level.

### Mathematical modelling
We implemented a multi-strain model to reconstruct the transmission dynamics of the dominant variants carrying the A220V and M234I-A376T mutations, all other co-circulating variants, and the Alpha variant (B.1.1.7) for Veneto and the rest of Italy between May 2020 and 2021. The month prior to the detection of A220V in GISAID in Italy was chosen as the start of the modelling period. We refer to A220V, Alpha and other variants as the concordant variants and to the M234I-A376T variant as the discordant variant (Supplementary Table 5).

**Epidemiological and genomic data**. Epidemiological and genomic sequencing data recorded in Veneto and the rest of Italy were used to reconstruct variant-specific incidences. Let subscript *Y* refer to the virus variant (*M* = M234I-A376T, *A* = A220V, *Al* = Alpha, *O* = other) and subscript *m* refer to the month. The number of sequences tested $n_m$ and the number of variant *Y* sequences identified $x_{Ym}$ were obtained from the GISAID databank[29]. The mean daily incidence of SARS-CoV-2 reported for month *m*, $R_{TOTm}$ was obtained from the Civil Protection[25]. Assuming that positive molecular samples are randomly selected for genomic sequencing, the monthly prevalence of the sequences from GISAID reflects the true monthly incidence of infection of each variant. From the data, we reconstruct the variant-specific reported incidence of infection $NR_{Ym}$ (Fig. 3c–f) by multiplying the daily mean incidence reported in month *m* (Fig. 3b) by the reported prevalence of variant *Y* in the same month (Figs. 3g–4j), $NR_{Ym} = R_{TOTm}\frac{x_{Ym}}{n_m}$, which is proportional to the true incidence of variant-specific infection.

Let subscript *i* refer to time, in days and subscript *T* refer to the test (*AN* = antigen, *MO* = molecular). To reconstruct the testing and reporting process in Italy (Fig. 4b), we obtained data on the number

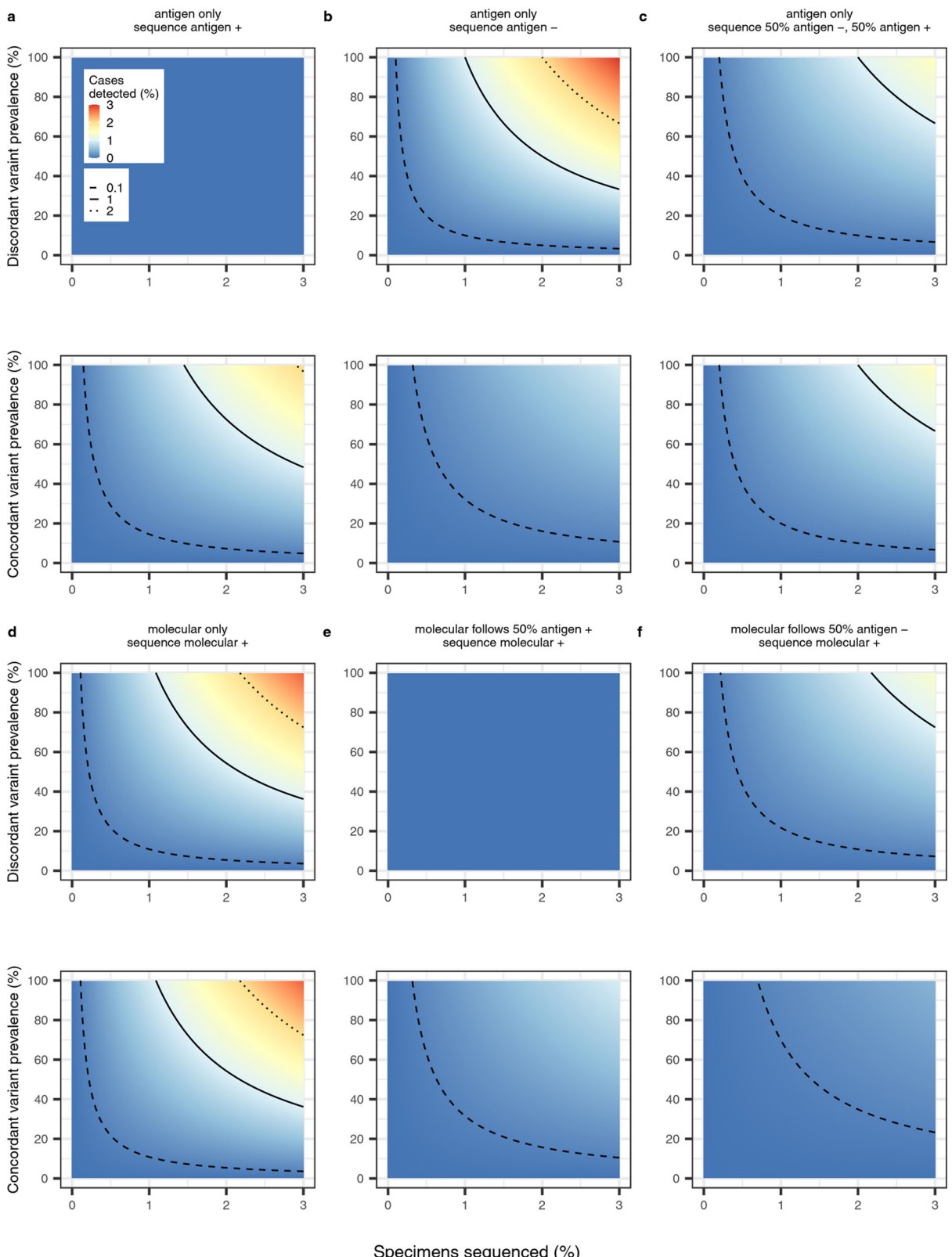

tests $N_{Ti}$ conducted in day $i$ from the Civil Protection[25]. Regional data on $N_{MOi}$ are published for the full modelling study period, whereas $N_{ANi}$ data are available from 16 January 2021 onwards only. In Veneto, the total number of antigen tests performed each month $N_{ANm}$ have been published from September 2020 onwards, and we assumed no antigen testing was conducted prior to this. To obtain an estimate of $N_{ANi}$ in Veneto between September 2020 and January 2021, we calculated the mean daily number of tests performed each month and linearly interpolated the missing values (Fig. 3a). In the rest of Italy, no data on antigen testing are available prior to January 2021, although there is evidence of antigen testing from October 2020 onwards[52, 53]. We therefore assumed that antigen testing commenced in October 2020 in Italy and used linear interpolation to estimate $N_{ANi}$ between October 2020 and January 2021 (Fig. 3a).

**Fig. 7 | Genomic surveillance of a discordant variant and a concordant variant across different testing and sequencing strategies, given a range of prevalence values. a** Individuals present for diagnosis with an antigen test and a sample of positive specimens are sequenced. **b** Individuals present for diagnosis with an antigen test and a sample of negative specimens are sequenced. **c** Individuals present for diagnosis with an antigen test and a sample of both negative and positive specimens are sequenced. **d** Individuals presenting for diagnosis with a molecular test and a sample of positive specimens are sequenced. **e** Individuals presenting for diagnosis with an antigen test, 50% of positive cases are confirmed with molecular and a sample of positive molecular specimens are sequenced. **f** Individuals presenting for diagnosis with an antigen test, 50% of negative cases are confirmed with molecular and a sample of positive molecular specimens are sequenced. Prevalence values range from 0 to 100% for both the discordant and concordant variants. Percentage of specimens sequenced ranges from 0 to 3%. Molecular sensitivity and specificity are 92% and 100%, respectively, regardless of the infecting variant. Antigen sensitivity is 68.9% against the concordant variant and 0% against the discordant variant. Dashed contour denotes 0.1%, solid contour denotes 1% and dotted contour denotes 2%.

To account for the SARS-CoV-2 vaccination campaign which began in December 2020, we obtained data on the mean daily number of second doses administered in Veneto and the rest of Italy in month $m$ $V_m$ from the Extraordinary Commissioner for the COVID-19 emergency[54].

**Multi-strain transmission model.** Supplementary Fig. 7 shows the flow diagram of the multi-strain susceptible (S), exposed (E), pre-symptomatic infectious (IP), asymptomatic infections (IA), symptomatic infectious (IS), quarantined (Q), recovered (R) compartmental model. We assumed a total population size $N = 4,847,026$ for Veneto and $N = 54,410,540$ for the rest of Italy. In accordance with the results reported in the national seroprevalence survey in July 2020[55], we assumed that 98% of the population of Veneto and 97% of the population of the rest of Italy were susceptible to SARS-CoV-2 at the start of the modelling period (i.e., were in the S compartment, $S_0 = 4,753,625$ for Veneto and $S_0 = 53,021,564$ for Italy). We assumed an average latency period $1/\eta$ of $1/1.31$ days[56], pre-symptomatic infectious period $1/\sigma$ of $1/3.79$ days[28] and an average infectious period $1/\gamma$ of 2.1 days[28]. We assumed the proportion of individuals developing symptoms $\mu$ to be 59%[28] and that only symptomatic individuals test and isolate (Q compartment).

We defined a variant-specific force of infection (i.e., the daily per capita risk that a susceptible individual becomes infected) $\lambda_{Yi}$, given by

$$\lambda_{Yi} = \frac{\beta_Y(IP_{Yi} + IA_{Yi} + IS_{Yi})}{S_0} \qquad (1)$$

Thus, $\lambda_{Yi}$ is proportional to the transmission rate of variant $Y$ ($\beta_Y$) and to the proportion of undetected infectious individuals of variant $Y$ on day $i$ (($IP_{Yi} + IA_{Yi} + IS_{Yi})/S_0$).

We assumed the molecular sensitivity $\phi_{MO}$ to be 92.0%[57] against both concordant and discordant variants and the antigen test sensitivity $\phi_{AN}$ to be 68.9% against the concordant variants and 0% against the discordant variant (Supplementary Table 3). The antigen and molecular test specificity $\varepsilon_T$ was assumed to be 100% against all variants. We estimated the proportion of symptomatic individuals infected with the discordant variant M234I-A376T detected by surveillance as:

$$\delta_{Mi} = \rho\, p_{MOi}\, \phi_{MO} \qquad (2)$$

Where $\rho$ is the probability of an infectious individual with symptoms testing (i.e., taking a diagnostic test) and $p_{MOi}$ represents the proportion of molecular tests that were conducted for diagnostic purposes at the point of testing (i.e., not to confirm the result of a positive antigen test) (Fig. 4b). The proportion of symptomatic individuals infected with a concordant variant that were detected and reported $\delta_{Yi}$ ($Y = A, O\, Al$) is given by:

$$\delta_{Yi} = \rho(\phi_{MO}\, p_{MOi} + \phi_{AN}(1 - p_{MOi})) \qquad (3)$$

where $1 - p_{MOi}$ is the proportion of antigen tests conducted. We assumed that detected symptomatic infections fully complied with isolation (Q compartment) and thus did not contribute to the onward transmission of the virus.

Let $p_{c,i}$ denote the per capita probability of being infected with a concordant variant:

$$p_{c,i} = \frac{\eta E_{Ai} + \eta E_{Oi} + \eta E_{Ali}}{S_0} \qquad (4)$$

Where $\eta E_{Yi}$ is the true incidence of infection of variant $Y$. $N_{ANi}\, p_{ci}$ is therefore the number of antigen-positive tests on day $i$. Thus, the probability of receiving a molecular test for diagnostic purposes (i.e., not for confirming an antigen test) $p_{MOi}$ is given by:

$$p_{MOi} = \frac{N_{MOi} - N_{ANi}\, p_{c,i}}{N_{MOi} + N_{ANi}(1 - p_{c,i})} \qquad (5)$$

The reproduction number of each variant ($R_{0Y}^0$) is given by:

$$R_{0Y}^0 = \frac{\beta_Y}{\sigma} + \frac{(1 - \mu)\beta_Y}{\gamma} + \frac{(1 - \delta_Y)\mu\beta_Y}{\gamma} \qquad (6)$$

To reflect the nationwide restrictions introduced in Italy in November 2020 and March 2021 in response to the second and third waves, respectively, we assumed that the reproduction number of all variants at the start of the modelling period was reduced by a factor $(1 - \omega_l)$ from 15 November 2020 to 14 March ($l = 1$) and reduced by a factor $(1 - \omega_l)$ from 15 March 2021 onwards ($l = 2$), for both Veneto and the rest of Italy:

$$R_{0Y}^l = (1 - \omega_l)R_{0Y}^0 \qquad (7)$$

where $\omega_l$ is the per cent reduction in transmission due to the implementation of interventions. To account for the national rollout of vaccination from January 2021 onwards, we assumed that a proportion of susceptible individuals acquired immunity and entered the recovered compartment through vaccination. The per capita mean daily rate of vaccination for month $m$ $v_m$ is given by:

$$v_m = \frac{V_m}{S_0} \qquad (8)$$

From the model, the daily mean reported incidence for variant Y in month $m$ is given by $NR_{Ym} = \delta_{Ym}\mu\sigma IP_{Ym}$. By equating this with $R_{TOTm}\frac{x_{Ym}}{n_m}$ (i.e., the available epidemiological and genomic data) and assuming that the number of sequences detected by genomic surveillance $x_{Ym}$ follows a Negative Binomial distribution, the likelihood of the data is given by:

$$x_{Ym} \sim \text{NegativeBinomial}\left(\frac{\delta_{Ym}\mu\sigma IP_{Ym}}{R_{TOTm}}n_m, k\right) \qquad (9)$$

where $\frac{\delta_{Ym}\sigma\mu IP_{Ym}}{R_{TOTm}}$ represents the probability that a sequenced sample test positive for the $Y$ variant in month $m$ and $k$ represents the over-dispersion parameter.

**Inferential framework.** The model was fitted to the observed data in a Bayesian Markov chain Monte Carlo framework using the No-U-Turn sampler via Rstan (version 2.21.0)[58] in R (version 4.1.1)[59] and RStudio[60] (version 2021.09.0), using the observed number of SARS-CoV-2 cases

($R_{TOTm}$), the observed number of molecular and antigen tests administered ($N_{MOm}$, $N_{ANm}$), the observed number of second vaccine doses administered ($V_m$), the number of sequences tested ($n_m$) and the number of sequences testing positive by variant ($x_{Ym}$) as input data. We simultaneously calibrate the model to the data from Veneto and the rest of Italy by fitting to the variant-specific reported incidence relative to the population size (i.e., divided by $S_0$ and using a scaling factor of 100,000), assuming the same transmission parameters $\beta_Y$, probability of a symptomatic individual taking a test $\rho$, and reduction in transmission $\omega_l$ across the two locations. We assumed that transmission of each variant started one month prior to its first detection in Italy in the GISAID databank. The size of the initial seed for each variant $I0_Y$, the probability of testing $\rho$, the transmission parameters $\beta_Y$ and the reduction in transmission $\omega_l$ were estimated, assuming the priors: $\beta_Y \sim$ normal(1,1), $\rho \sim$ beta(1,1), $\omega_l \sim$ beta(1,1), $I0_Y \sim$ normal(1,10) for Veneto and $I0_Y \sim$ normal(1,100) for Italy. We additionally estimated the overdispersion parameter $k$, assuming the prior $k \sim$ exponential(0.01).

To improve the fit, we weighted the likelihood of the M234I-A376T variant by 10. We simulated 4 chains of 2,000 iterations each, discarding the first 1000 iterations and assessed convergence using the R-hat convergence diagnostic. 95% CrI of estimates were obtained from the 2.5 and 97.5 percentiles of 100 samples of the posterior distributions. All data and code used to fit the model are available on GitHub[61].

**Sensitivity analysis.** We explored four different testing assumptions and compared their fit using the log-likelihood. All four model variants had comparable log-likelihoods so we chose the visually best fitting model as the baseline model. This model assumed that only symptomatic individuals undergo diagnostic testing and that they isolate if they receive a positive result, which is in agreement with the Italian testing policy during the modelling study period. In a second scenario, we assumed that all symptomatic individuals isolate and that asymptomatic individuals test with a probability $\rho$ and isolate if they receive a positive result. In the third scenario, we assumed that symptomatic individuals test and isolate if they receive a positive result, and that asymptomatic individuals test with a probability $\rho$ and isolate if they receive a positive result. In a fourth scenario, we assumed that the probability of taking a diagnostic test is independent of symptom occurrence, and that asymptomatic and symptomatic individuals test with the same probability $\rho$ and isolate if they receive a positive result.

We also ran a thorough sensitivity analysis on the main model. Firstly, we assumed that symptomatic infectious individuals who did not test or who receive a false negative test result limit their contacts by a factor $(1 - \alpha)$, which was estimated from the data assuming the prior $\alpha \sim$ beta(1,1). Secondly, we assessed the robustness of the results when assuming antigen test sensitivities of 64.3% and 87.5% (Supplementary Tables 3 and 4) against the concordant variant. Thirdly, we explored the sensitivity of the results on the assumptions made about the average latency period, pre-symptomatic infectious period and infectious period which were respectively fixed to (i) 1.31 days[28], 4.29 days[62] and 1.6 days[28] and (ii) 2.15 days[28], 2.95 days[56] and 2.1 days[28]. Finally, we evaluated the impact of assuming 100% vaccine efficacy $\zeta$ against infection following two vaccine doses by exploring the cases $\zeta$ = 70% and 80% against infection[63].

**Counterfactual analysis.** In a separate analysis, we explored four counterfactual molecular and antigen testing scenarios in Veneto. In scenario 1, we assumed that the proportion of antigen and molecular tests conducted in Veneto was the same as in the rest of Italy over the study period. In scenario 2, we assumed that a molecular test followed a negative antigenic test and that those receiving a positive antigenic test isolate without molecular confirmation. Finally, we modelled an antigen-only testing strategy (without molecular confirmation), assuming antigen sensitivity against the concordant variants to be

68.9% (Scenario 3) and 87.5% (Scenario 4), respectively (Supplementary Tables 3 and 4).

Finally, in a separate analysis, we explored the transmission dynamics of M234I-A376T assuming different values of $R_{OM}$ compared to the fitted estimates, which were obtained by multiplying the estimated $R_{OM}$ value by 0.8, 1.2, 1.4 and 1.6.

**Estimating test performance metrics.** We calculated the *PPV*, *NPV* and $p(T^+)$ for all possible combinations of concordant $\theta_c$ and discordant $\theta_d$ prevalence values under different testing strategies. Let $\phi_{Tc}$ and $\phi_{Td}$ denote the test sensitivity against the concordant and discordant variants, respectively. Let $\varepsilon_T$ denote test specificity, as above. Note that for molecular tests $\phi_{MOc} = \phi_{MOd}$. We define the true positive ($TP_T$), true negative ($TN_T$), false positive ($FP_T$) and false negative ($FN_T$) values are as follows:

$$TP_T = \phi_{Tc}\theta_c + \phi_{Td}\theta_d \quad (10)$$

$$TN_T = \varepsilon_T(1 - \theta_c - \theta_d) \quad (11)$$

$$FP_T = (1 - \varepsilon_T)(1 - \theta_c - \theta_d) \quad (12)$$

$$FN_T = (1 - \phi_{Tc})\theta_c + (1 - \phi_{Td})\theta_d \quad (13)$$

If only molecular or only antigen testing is conducted, then:

$$PPV_T = \frac{TP_T}{TP_T + FP_T} \quad (14)$$

$$NPV_T = \frac{TN_T}{TN_T + FN_T} \quad (15)$$

$$p(T_T^+) = TP_T + FP_T \quad (16)$$

In testing strategies using both molecular and antigen testing, we assume individuals presenting for diagnosis initially receive an antigen test and that $X$ proportion of either positive or negative antigen tests are followed up with a molecular test. We assume that if there is a discordant result between the antigen and molecular tests, the molecular test result is reported, and that concordant antigen and molecular results are only reported once. Thus, if a proportion $X$ of antigen-positive tests are confirmed with molecular, we obtain:

$$PPV = \frac{[TP_{AN} \, X \, \phi_{MO}] + [(1-X)TP_{AN}]}{[TP_{AN} \, X \, \phi_{MO}] + [(1-X)TP_{AN}] + [FP_{AN}X(1-\varepsilon_{MO})] + [(1-X)FP_{AN}]} \quad (17)$$

$$NPV = \frac{TN_{AN} + [FP_{AN} \, X \, \varepsilon_{MO}]}{TN_{AN} + [FP_{AN} \, X \, \varepsilon_{MO}] + FN_{AN} + [TP_{AN} \, X \, (1-\phi_{MO})]} \quad (18)$$

$$p(T^+) = [TP_{AN} \, X \, \phi_{MO}] + [(1-X)TP_{AN}] + [FP_{AN}X(1-\varepsilon_{MO})] + [(1-X)FP_{AN}] \quad (19)$$

$$\% \text{ of diagnostic tests that are antigen} = \frac{1}{1 + X(TP_{AN} + FP_{AN})} \quad (20)$$

Similarly, if a proportion $X$ of antigen-negative tests are confirmed with a molecular test, we obtain:

$$PPV = \frac{TP_{AN} + [FN_A \, X \, \phi_{MO}]}{TP_{AN} + [FN_{AN} \, X \, \phi_{MO}] + FP_{AN} + [TN_{AN} \, X \, (1-\varepsilon_{MO})]} \quad (21)$$

$$NPV = \frac{[(1-X)TN_{AN}] + [TN_{AN} \, X \, \varepsilon_{MO}]}{[(1-X)TN_{AN}] + [TN_{AN} \, X \, \varepsilon_{MO}] + [(1-X)FN_{AN}] + [FN_{AN} \, X \, (1-\phi_{MO})]} \quad (22)$$

$$p(T^+) = TP_{AN} + [FN_{AN} \, X \, \phi_{MO}] + FP_{AN} + [TN_{AN} \, X \, (1 - \varepsilon_{MO})] \quad (23)$$

$$\% \text{ of diagnostic tests that are antigen} = \frac{1}{1 + X(TN_{AN} + FN_{AN})} \quad (24)$$

In this analysis, we assumed that the sensitivity and specificity of molecular tests are 92%[57] and 100%, respectively, regardless of the infecting variant and that antigen sensitivity was 68.9% against the concordant variant and 0% against the discordant variant (Supplementary Table 3). We allowed for imperfect antigen specificity (99.68%) against both variants[64] to explore the variability in PPV (i.e., there is no variability in PPV assuming 100% specificity).

**Estimating detection by genomic surveillance.** Let $\varphi$ denote the proportion of viral samples sequenced for genomic surveillance (ranging from 0 to 3%)[37] and $P_{Vseq}$ denote the percentage of cases of a discordant or concordant variant (denoted subscript V) that are detected by genomic surveillance. If only molecular or only antigen testing is conducted and positive samples are sequenced then:

$$P_{Vseq} = \phi_{T_V} \, \theta_V \, \varphi \quad (25)$$

Equally, if only molecular or only antigen testing is conducted and negative samples are sequenced then:

$$P_{Vseq} = \left(1 - \phi_{T_V}\right) \theta_V \, \varphi \quad (26)$$

If a proportion $X$ of antigen-positive tests are confirmed with a molecular test and positive molecular samples are sequenced, then:

$$P_{Vseq} = \phi_{AN_V} \, \theta_V \, X \, \phi_{MO_V} \, \varphi \quad (27)$$

If a proportion $X$ of antigen-negative tests are confirmed with a molecular test and positive molecular samples are sequenced, then:

$$P_{Vseq} = \left(1 - \phi_{AN_V}\right) \theta_V \, X \, \phi_{MO_V} \, \varphi \quad (28)$$

**Reporting summary**
Further information on research design is available in the Nature Research Reporting Summary linked to this article.

## Data availability
The SARS-CoV-2 genomic sequences of the concordant and the discordant samples are available in Supplementary Data 1, together with their relative GenBank and GISAID accession numbers. The GISAID acknowledgements table is provided as Supplementary Data 2, where all the GISAID accession numbers of the sequences downloaded from GISAID for this study are available. All data used in the modelling study are on GitHub at https://github.com/bnc19/COV_Italy_multistrain.

## Code availability
Code is available at: https://github.com/bnc19/COV_Italy_multistrain [61].

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

## Acknowledgements

We gratefully acknowledge the authors from the Originating Laboratories and the Submitting Laboratories who generated and shared via GISAID the data on which this research is based (see Supplementary Data 2). The work was funded by the MRC Centre for Global Infectious Disease Analysis (reference MR/R015600/1), jointly funded by the UK Medical Research Council (MRC) and the UK Foreign, Commonwealth & Development Office (FCDO), under the MRC/FCDO Concordat

agreement and is also part of the EDCTP2 programme supported by the European Union; and acknowledges funding by Community Jameel. B.C.D. discloses support for the research of this work from the MRC Doctoral Training Partnership. I.D. discloses support for the publication of this work from the Royal Society and Wellcome Trust (grant number: 213494/Z/18/Z). S.T. discloses support for the research of this work from the COVID-19 emergency fund from the University of Padova (grant number: TOPP_PRIV20_01).

## Author contributions

C.D.V., A.C., and S.T. developed the hospital study protocol. I.D. conceived the modelling. G.B., C.D.V., F.O., E.F., A.M.C., and V.C. collected the data and organised the experimental protocols. A.R.B., S.T., F.B., E.L., and L.M. analysed the data. B.C.D., C.C., and I.D. developed the mathematical model. B.C.D. performed the modelling analysis and wrote the first draft of the manuscript, with contributions from I.D. and S.T. I.D., S.T., and A.C. jointly supervised the work and contributed equally to this manuscript. C.D.V. and B.C.D. are joint first authors and contributed equally to this manuscript. All authors revised the manuscript critically and approved the final version.

## Competing interests

The authors declare no competing interests.

## Ethical approval

The data were extracted from the electronic medical record administrative database (Galileo platform) of the University Hospital of Padova. The purpose and design of the study has been approved by the Local Ethics Committee of the Province of Padova (Italy) (Protocol Numbers: 69295, 0001609). The need for written informed consent was waived for patients because this is a retrospective study and this is in accordance with the Italian Drug Agency note 20 – March 2008 (GU Serie Generale no. 76 31/3/2008).
