## [Peer Review File · Nature Communications]

Impact of antigen test target failure and testing strategies on the transmission of SARS-CoV-2 variantsREVIEWER COMMENTS

Reviewer #1 (Remarks to the Author):

In this paper, the authors identify a variant that circulated in late 2020 and early 2021 in Veneto, Italy that was not detectable by antigen tests. They then develop a mathematical model that suggests that the epidemic and the distribution of the circulating types was strongly affected by the relative level of antigen testing in the region. The identification of sequences that can evade antigen tests, which continue to form the backbone for national testing policies in many countries, is clearly important. However, I have some concerns with the model.

The overall narrative from the model is that due to the 0% sensitivity of antigen tests for the M234I-A376T variant and 64.3% against other variants, the M234I-A376T variant was able to spread more easily. Other places in Italy that used more PCR testing were able to detect more M234I-A376T and therefore could curtail some of its spread. The model relies on everybody who tests positive, self-isolating and not transmitting to anyone and therefore everybody who gets infected with M234I-A376T and gets an antigen test goes on to transmit. However, this is an extreme scenario, that ignores people limiting their exposures due to feeling unwell. It also ignores the timing of testing since infection, and that people may have done their transmitting prior to testing – this was particularly relevant in this stage of the epidemic, with substantial pre-symptomatic transmissibility.

Relatedly, the sensitivity of 64% for other variants is probably much higher when considering time periods when individuals are at risk of transmission.

Several of the model outputs are confusing. In particular, the underlying betas (transmission rates) for the alpha, A220V and 'other' variants is twice as high in Veneto as the rest of Italy, however the M234I-A376T transmission rate is approximately the same across the two locations. In addition, 76% of infected individuals got tested in Veneto compared to 29% in the rest of Italy. I found it difficult to come up with viable hypotheses to explain such discrepancies –that the difference in transmissibility between locations should differ by variants (especially for the key variant we are interested in), that there were such substantial differences in accessibility to tests by the end of 2020. I suspect that some of these slightly unexpected estimates may be due to the extreme modelled scenario of complete control of detected cases.

The authors use different ranges when plotting the epidemic curves in Veneto and the rest of Italy, which hinders comparison in Figure 5A, but it seems that the timing of the first epidemic peak is modelled to be earlier in Veneto than the rest of the country. It would be useful to compare that with reported epidemiological data to see if that was indeed the case.

There is no reference to time since symptom onset when calculating the sensitivity of the tests to the different variants. Individuals can stay PCR positive (which the authors use as the gold standard) for a long time after individuals test negative on antigen tests. As the authors point out, antigen tests are a good marker of viable virus and linked to transmissibility – therefore reduced sensitivity after e.g., 10 days may not be important for transmission. The authors should set out the time since symptom onset for the variant results.

Overall, the weaknesses in the model detract from the paper. At the very least, I would plot the observed proportion that were of each variant in 5C and not just rely on the model outputs. This would provide much more direct evidence of a difference on the epidemic trajectories from the antigen testing sensitivity. Secondly, I would see if there was a difference in proportions of each variant between older and younger individuals in Veneto – as younger individuals all got PCR tested, you should see a reduced proportion of M234I-A376T in the younger age groups than older ones (assuming age-dependent mixing).

Reviewer #2 (Remarks to the Author):

Del Vecchio and colleagues analyse the emergence and spread of a SARS-CoV-2 variant escaping antigenic testing in Italy. The reconstruction of the variant spread in the Veneto region, compared to the whole Italy, joint with the analysis of counterfactual scenarios show how the massive use of antigenic testing could favour the emergence of escaping variants. This unintended consequence of antigenic testing calls for the design of testing strategies that combine in a reasoned way antigenic and molecular testing.

The study has important public health implications. The frequent emergence of Variants of Concern has shown the importance of molecular testing and full genome sequencing to monitor virus circulation. This work adds an important and original point to the discussion by unveiling the risk of a testing strategy too centred on antigenic testing.

The analysis presented in the manuscript is extensive and complete. To the best of my understanding it is sound and provides a strong support to authors' hypothesis. Still, some improvements are needed before I can recommend the acceptance of the manuscript.

Overall the manuscript reads well. However, the presentation of the work is not always easy to follow, because the analysis is rich and the manuscript dense. I believe authors should make an extra effort to make the study more accessible to a wider audience. The brief synthesis presented at the end of the Introduction section could be extended to better anticipate the different parts of the analysis and guide the reader throughout the results. The different subsections of the Results section could be better linked one to the other, e.g. with sentences placed at the beginning of subsections to anticipate their aim. I also have specific remarks on the presentation of the methods:

- 1) the different parts of the analysis refer to different study periods. However, study periods are not always clearly indicated in the text
- 2) Authors write in the main text "the temporal dynamics of the reproductive number of each variant are described in Supplementary Figure 7". From this sentence I understand that the figure shows the R_t index – as can be computed with the Cori et al method. However, from a close look to the figure this does not seem the case and the quantity plotted seems, instead, the one defined in Eq. (6). Authors should be clearer on this
- 3) is ω_2 (the percent of reduction in transmission after control measures for the third wave) applied to the initial transmissibility or to the transmissibility already reduced by ω_1 ?

Figures should be improved. In particular Figure 4 is not easy to read – especially panels b-e. Curves in panels a and b of Figure 5 are not clearly visible. In general, readability of figures should be improved by adding legends and panels' titles directly to the figures when missing – e.g. description of line styles in Figure 6; legends and panels titles in Figure S4.

Authors provide some sensitivity analysis to the multi-strain model. Still, the model relies on the choice of several parameters. I believe authors should consider expanding the sensitivity analysis (e.g. incubation and infection period, effectiveness vaccination) to show that their results are robust to different modelling assumptions.

The Discussion is well articulated. However, limitations are missing. One limitation that should be discussed is the fact that sequencing is not always aleatory and certain viruses are prioritised over others – based for instance on molecular testing characteristics, as authors note. This could bias the derivation of variant-specific incidence – although I suspect this to affect mainly the incidence of the Alpha variant.

Minor revisions:

"Some of the supernatants were evaluated with.": This sentence is misplaced or unclear

"Sequence data and corresponding metadata, released from 17th of February 2021": Should it be 2020?

"We obtained data on the mean daily number of second doses administered in Veneto and the rest

of Italy ...": No reference to vaccination was done up to this point. As such the sentence is confusing.

"We assumed that 98% and 97% of the population of Veneto and the rest of Italy were susceptible to SARS-CoV-2 as of July and May 2020": Maybe July and May were inverted?

"The monthly prevalence trend of the variants found in concordant and discordant cases was plotted from January to December 2020 and reported for Europe, Italy, and the Veneto region (Supplementary Figure S5).": Only Italy and the Veneto region are reported in Figure S5.

Reviewer #3 (Remarks to the Author):

Thank you for the opportunity to review this manuscript. The reporting of novel COVID-19 variants circulating within Veneto that are not detected by current rapid tests is an important and highly relevant finding. The authors produced a mathematical model to illustrate how various testing strategies may select for these novel variants, the results of which can be used to help inform policy.

Overall, I found the introduction well written, providing the necessary context for the study. The results were detailed, but sometimes difficult to understand, without first reviewing the methods and supplementary material. The discussion was again well written, but could have included additional information (see comments below).

Further specific comments:

1. Methods (study design) and results (lines 103-110). The study design states that all subjects were tested with both antigen and molecular assays. However, line 314 states that individuals from 0-19 years were predominantly tested with molecular swab, with few tested by both assays. I do not understand; if the study design was to test with both antigens, why did this not apply to a certain age group? The study design section implies that the sampling protocol was specific for this study (eg "Swab samples were collected from 15th September 2020 to 16th October 2020 from patients ..." and " All subjects were tested within one hour with both antigen..." and "Information was collected from each individual about their age, sex, date of sampling, symptoms..."). However the ethics statement reads "The data were extracted from the electronic medical record administrative database". The authors should clearly state whether the data were prospectively collected so that they had control over the test protocol, or whether the data were extracted from routinely collected clinical data.
2. Methods (line 323). It is not clear how the results from the in-house real-time RT-PCR informed the study. Were samples that had a single positive target on the molecular assay, but a positive RT-PCR result classified as positive? Or were only samples with a positive result for both viral targets included, irrespective the of the RT-PCR testing?
3. The transmission model structure (Figure S6) does not include any infection of susceptibles based on the infectious reservoir. Therefore, I am unclear whether the results of the model (in terms of number of infectious individuals) informs subsequent time steps, or whether it is simply an output to compare against epidemiological data. If there is no feedback within the model then I am not sure why recovered compartment is included since the rate of recovery is assumed.
4. Results (lines 142 & 143). The reported sensitivity values should have 95% CIs reported. Only 3 samples contained the M234I and A37T mutations, which is insufficient to make any definitive statement about test performance against this variant. The results presented also seem open to the effects of confounding and bias. Figure S3 clearly shows the test sensitivity decreases with increasing Ct, and the authors have found that discordant samples have significantly higher Ct values. In addition, the authors have elected to sample 8 discordant and 9 concordant samples, dramatically changing the sample profile where discordant samples only made up 1% of all samples. For all of these reasons it is likely the reported sensitivity of 64.3% against variants not carrying the substitution is under-estimated. The value of 64.3% sensitivity is used in the

modelling for concordant variants. I am wondering why the authors have elected to use this value, which is based on only a relatively small number of tests, rather than the more robust 68.9% obtained for all samples in the hospital study.

5. There seems to be a conflict in the reporting of the statistical test used to compare the Ct values between discordant and concordant samples. The methods and Table S2 state Wilcoxon-Mann-Whitney two-sample test was used, but the results report a Welch two-sample t-test. If the Mann-Whitney was used then it would be more appropriate to report the median values in Table S2, rather than the mean.

6. Methods (line 429). The proportion of undetected infectious individuals on any day is calculated as the number in this category divided by S_0 . It is unclear to this reviewer why S_0 is used as the denominator, and not S_i since the susceptible pool would be expected to decrease over time.

7. Discussion. The discussion is well written but lacks any mention of model assumptions or limitations and how these may impact the results presented. For instance, how realistic is it that detected infected individuals do not contribute to any transmission; they may go into isolation after detection, but this does not mean they are not infectious prior to detection. It seems that double-vaccinated individuals are not able to become infected, which is clearly not the case. Would changing these assumptions change the conclusions from the model? Another point of interest is that the fitted values for Veneto for many parameters are higher than for Italy, so it would be useful to understand whether there is any evidence to support this. For example, is there any evidence that the reduction in transmission after additional control measures was greater in Veneto than Italy overall?

Minor comments

8. Consistency in the use of terms. It would be helpful if the antigen and molecular tests were referred to in the same way through the manuscript. For instance, in the paragraph starting line 111 the following terms are used: antigen test, PanbioTM ANCOV test, ANCOV+, Abbott test.

9. Line 88: "... that can escape detection by antigen testing."

10. Figure 1. Wording of title does not seem to accurately reflect the content of the figure. Suggested amending to "Proportion of COVID-19 patients receiving antigenic diagnostic tests in Veneto...", or similar.

11. Figure 2. Final sentence "The under 20 subjects have not been considered in the study." is not grammatically correct. It is not immediately clear what information is added by separating the 'Emergency' and 'Infectious Disease' subjects in one part of the flow chart. This could be removed for improved clarity. If this element stays in the flow chart then I would suggest making sure the labelling of subjects aligns with the figure title (eg Emergency vs A&E).

12. Table 1. It is not clear what "P" refers to in the second row of the table. Please include abbreviation in title or footnote. I am assuming the authors are using P to represent the prevalence of disease, since both PPV and NPV depend on the prevalence. However, I do not see the relevance of varying estimates of disease in this Table; the proportion of PCR-positive patients will provide the prevalence of disease in this cohort, which dictates the test performance. From Figure 1 I have been able to calculate the prevalence of disease in this cohort as 0.044, which aligns with the first reported value of P in Table 1. But this information should be more obvious in the manuscript. Also, the first PPV and NPV values for all data are not displaying properly (first number missing).

13. Figure S5. G204R and R203K are listed as variants in concordant samples. While this is true, according to Figure 3 they were also seen in discordant samples. Suggest the authors include this information in the S5 figure title.

14. Figure 4. Please define GISAID.

15. Figure 5c. From the figure title and the information within the plot I am not certain of what this sub-figure is actually showing. "Percentage of A220V, M234I-A376T, alpha, other virus variants and total cumulative incidence reported in Veneto and the rest of Italy." This sounds like actual data, but the following sentence and data in plot indicate this is modelled output. The vertical axis is "Percent of cumulative incidence reported". At what time point is this cumulative percent? The figure legend has no sub-plot d) but includes sub-plot f), while figure only has a-e.

16. Table S1. Please include the number of samples used for each Ct threshold.

17. Some references seem incomplete and may need a more information like a url (eg reference 24).

18. Line 333. The incomplete sentence should be removed.

Reviewer #4 (Remarks to the Author):

This study reports detection of SARS-CoV-2 variants that escape rapid antigen tests in Veneto Italy, potentially via mutations in their N protein, highlighting the need for retaining PCR testing for virus surveillance. The manuscript is well written, and was a pleasure to read. The study appears apt and timely, especially considering its potential usefulness to scientists, public healthcare professionals and policy makers.

I have only one major comment and a few minor ones.

Major comment:

-Figure 3 is not clear. The mutations shaded in yellow and highlighted in red are present in both the ANCOV+/DNCOV+ and ANCOV-/DNCOV+ samples. Is this meant to show that they are present in both groups? If so, why is it that some of the other samples have these same mutations but they are not highlighted? For example all the samples appear to have the R203K and G204R mutations, but its only highlighted for the 1st (D1_7_B.1.1.119) and the 9th (C1_2_B.1.1.1) samples. The legend also says that "The last six sequences do not contain any mutation on N gene according to the Wuhan reference sequence", but they obviously do from the alignment shown. Also from the alignment shown, there doesn't seem to be mutations common to the ANCOV-/DNCOV group that are not also present in the ANCOV+/DNCOV+ group, and some of the sequences shown look identical (for example the 7th sample (D2_2_B.1.177) looks identical to the last sample (C_Vo_46_BB)). All of the samples, except one, contain the R209I mutation, etc. The text on page 5 says "only 2/9 of the viruses present in the concordant samples had mutations within the region of mapped B cell epitopes, and none showed amino-acid substitutions downstream to position 220", which is obviously not the case from the figure. Are there typos in the alignment, or am I missing something?

Minor comments:

-Having figure 1 referred to in the introduction is a bit unconventional, and I wonder if it will be better to move it to the results section instead, or perhaps to a supplementary figure

-It may be useful to state the RNA extraction method(s) used.

-From the manuscript, there is no indication that the authors included negative controls in the RNA extraction, library preparation or sequencing steps, and their criteria for handling detection of mapped reads in the negative controls.

-How did the authors handle patients that were sampled more than once for the same test type (ANCOV or DNCOV)? I think this is not clear from the manuscript. For example, were there

patients that had more than one ANCOV tests and then had DNCOV tests during the study period? Were these concordant/discordant? etc

-From the Accession numbers provided in the supplementary data, it looks like the authors have deposited the sequences to the GISAID database. It may be useful to explicitly state that this has been done in the manuscript, including the type of data deposited (consensus sequences, bam files, FASTQ files) etc.

-Typo on line 119: "as shown in Supplementary Figure S4" should read "is shown in Supplementary Figure S4"

-lines 186-187: "implemented in across Italy" should read "implemented across Italy"

-line 333 contains an incomplete statement: "Some of the supernatants were evaluated with."

-I imagine the 'TM' in "DNA-freeTM" on line 337 is meant to be a superscript

-line 371, statement unclear: "Sequence data and corresponding metadata, released from 17th of February 2021, were downloaded from GISAID databank". Are these samples deposited to GISAID by other labs. In that case 17th february to when (or are these the samples that were available on the 17th February)?

Impact of antigen test target failure and testing strategies on the transmission of SARS-CoV-2 variants

Claudia Del Vecchio, Bethan Cracknell Daniels, Giuseppina Brancaccio, Alessandra Rosalba Brazzale, Enrico Lavezzo, Constanze Ciavarella, Francesco Onelia, Elisa Franchin, Laura Manuto, Federico Bianca, Vito Cianci, Annamaria Cattelan, Ilaria Dorigatti, Stefano Toppo, Andrea Crisanti

Reviewer #1 (Remarks to the Author):

In this paper, the authors identify a variant that circulated in late 2020 and early 2021 in Veneto, Italy that was not detectable by antigen tests. They then develop a mathematical model that suggests that the epidemic and the distribution of the circulating types was strongly affected by the relative level of antigen testing in the region. The identification of sequences that can evade antigen tests, which continue to form the backbone for national testing policies in many countries, is clearly important. However, I have some concerns with the model.

We thank the reviewer for their comments and for pointing out their concerns on the model. Below we provide a response to each point listed.

Comments:

- 1. The overall narrative from the model is that due to the 0% sensitivity of antigen tests for the M234I-A376T variant and 64.3% against other variants, the M234I-A376T variant was able to spread more easily. Other places in Italy that used more PCR testing were able to detect more M234I-A376T and therefore could curtail some of its spread. The model relies on everybody who tests positive, self-isolating and not transmitting to anyone and therefore everybody who gets infected with M234I-A376T and gets an antigen test goes on to transmit. However, this is an extreme scenario, that ignores people limiting their exposures due to feeling unwell. It also ignores the timing of testing since infection, and that people may have done their transmitting prior to testing – this was particularly relevant in this stage of the epidemic, with substantial pre-symptomatic transmissibility.**

We thank the reviewer for highlighting limitations in the assumptions made the original model. We have now extended the model to include a pre-symptomatic infectious compartment, which allows to capture pre-symptomatic transmission as noted by the reviewer. We have also divided the infectious compartment into two sub-compartments, one for symptomatic and one for asymptomatic infections. Thus, we assume that individuals progress from the exposed compartment to the pre-symptomatic compartment at a rate of $\alpha = 1/1.31$ days, and from the pre-symptomatic compartment to the infectious compartments at a rate of $\sigma = 1/3.79$, parameterised from ¹. We assume that $\mu = 59\%$ of cases are symptomatic, also parameterised from ¹.

Given these model extensions, we then explored 4 different testing scenarios: i) only symptomatic individuals test with a probability of ρ and isolate if they receive a positive test result; ii) all symptomatic individuals test and isolate and asymptomatic individuals test and isolate, with a probability of ρ ; iii) all symptomatic individuals test and isolate if they receive a positive test result, asymptomatic individuals test with a probability ρ and isolate if they receive a positive test result; iv) symptomatic and asymptomatic individuals both test with a probability of ρ and isolate if they are positive. As before, we estimate ρ .

For each model, we assessed the fit visually and using the deviance information criterion (DIC). We found that model i) where we assume that only symptomatic individuals have access to diagnostic tests was the best fitting visually, and had the lowest DIC. This is in agreement with the testing policy in Italy at the time, where the majority of tests were only available to individuals who had symptoms, and asymptomatic testing only occurred if individuals were identified through contact tracing. Therefore, we chose model i) as the main model, and present the results of the other scenarios in the supplementary material.

Given the main model, we also explore a model where symptomatic individuals who do not test, or who receive a negative test result, limit their exposure by reducing their contribution to the force of infection λ_Y by a factor α (which we estimate), as follows:

$$\lambda_Y = \frac{\beta_Y (IP_Y + IA_Y + (\alpha \cdot IS_{Yi}))}{S_0}$$

where β_Y is the variant specific transmission rate, IP_Y , IA_Y and IS_Y are the pre-symptomatic, asymptomatic and symptomatic infectious compartments respectively.

We thank the reviewer for the constructive comments which have allowed us to capture pre-symptomatic transmission and delays in the timing of testing since the onset of the infectious period, differences in symptom occurrence and to test alternative testing scenarios. The best model provides evidence that most symptomatic individuals do test and isolate (probability of symptomatic testing 92.7%). Given this, the model where we explore the extent to which symptomatic cases limit their exposure, independently of testing, estimated only a limited reduction in symptomatic transmission (9.28%; 95% CrI, 0.26-30.19). We therefore include this model as a sensitivity analysis, rather than as the baseline model.

Summary of changes made:

- We extended the transmission model to include a pre-symptomatic, asymptomatic and symptomatic infectious compartments, in place of a single infectious compartment as in the original version of the paper.
- We re-ran the analysis assuming 4 different testing scenarios to test hypotheses about the testing policy implemented. We updated Figure 5, Supplementary Figure 6, Supplementary Table 6 and Table 2 accordingly.
- We include a simplified flow diagram of the transmission model in Figure 4a and provide the full version of the model in SI Figure S6.

- We updated the figure legend of Figure 4b to reflect that only symptomatic individuals present for testing in the main model: *“(1) Symptomatic individuals present for diagnosis with either an antigen test or molecular test.”*
- In lines 221-223 of the results we edited the text as follows: *“We estimated the probability of a symptomatic infectious individual taking a diagnostic test to be 92.7% (95% CrI, 72.8-99.8%) for Veneto and Italy.”*
- In lines 205-209 of the results we edited the text as follows to introduce the 4 models of population testing explored: *“We fit 4 different model variants of population testing, varying assumptions on access to testing (symptomatic vs. asymptomatic cases), as well as the probability of isolating given a false negative result (See Methods for a full description). Supplementary Table S6 shows the posterior mean, 95% CrI and DIC (Deviance Information Criterion) of each model. The best fitting model assumed that only symptomatic cases test and isolate, and is thus presented as the main analysis.”*
- In the results lines 230 to 238 we added the following sentence: *“In a sensitivity analysis we assessed the impact of assuming lower and higher estimates of antigen sensitivity against the concordant variants; different infectious and latency periods; imperfect vaccine efficacy against infection and the extent to which symptomatic individuals limit their exposure, independently of testing. Across all scenarios and all parameters we found no significant difference in the posterior mean and 95% CrI (Supplementary Table S7).”*
- In the methods in lines 539 to 541 we edited the description of the model compartmental as follows: *“Supplementary Figure S6 shows the flow diagram of the multi-strain susceptible (S), exposed (E), pre-symptomatic infectious (IP), asymptomatic infectious (IA), symptomatic infectious (IS), quarantined (Q), recovered (R) compartmental model.”*
- In the methods in lines 547 to 550 we updated the following lines describing of the transmission model: *“We assumed an average latency period $1/\eta$ of $1/1.31$ days², pre-symptomatic infectious period $1/\sigma$ of $1/3.79$ days¹ and an average infectious period $1/\gamma$ of 2.1 days^{1,1}. We assumed the proportion of individuals developing symptoms μ to be 59% ¹ and that only symptomatic individuals test and isolate (Q compartment).”*
- In lines 551 to 554 of the methods we edited equation 1, defining the force of infection: *We defined a variant-specific force of infection (i.e., the daily per-capita risk that a susceptible individual becomes infected) λ_{Yi} , given by*

$$\lambda_{Yi} = \frac{\beta_Y (IP_{Yi} + IA_{Yi} + IS_{Yi})}{S_0} \quad \text{Eq. (1)}$$

Thus, λ_{Yi} is proportional to the transmission rate of variant Y (β_Y) and to the proportion of undetected infectious individuals of variant Y on day i ($(IP_{Yi} + IA_{Yi} + IS_{Yi})/S_0$). ”

- In the methods lines 561 to 562 we updated the text as follows to reflect that the probability of taking a diagnostic test ρ applies only to symptomatic individuals: *“Where ρ is the probability of an infectious individual with symptoms testing (i.e., taking a diagnostic test)”*

- We updated the methods in lines 564 to 568 to reflect that testing and isolation only applies to symptomatic individuals: *“The proportion of symptomatic individuals infected with a concordant variant that were detected and reported δ_{Yi} ($Y = A, O, AI$) is given by*

$$\delta_{Yi} = \rho(\phi_{DN} p_{MOi} + \phi_{AN}(1 - p_{MOi})) \quad \text{Eq. (3)}$$

where $1 - p_{MOi}$ is the proportion of antigen tests conducted. We assumed that detected symptomatic infections fully complied with isolation (Q compartment) and thus did not contribute to the onward transmission of the virus.”

- In the methods lines 569 to 571, we updated equation 4 to state that the true incidence of infection is now the rate of individuals entering the pre-symptomatic infectious compartment: *“Let $p_{c,i}$ denote the per capita probability of being infected with a concordant variant:*

$$p_{c,i} = \frac{\eta E_{Ai} + \eta E_{Oi} + \eta E_{Aii}}{S_0} \quad \text{Eq. (4)}$$

Where ηE_{Yi} is the true incidence of infection of variant Y .”

- In the methods in lines 574 we updated equation 6 which defines R_0 for the main model: *“The reproduction number of each variant (R_{0Y}^0) is given by:*

$$R_{0Y}^0 = \frac{\beta_Y}{\sigma} + \frac{(1-\mu)\beta_Y}{\gamma} + \frac{(1-\delta_Y)\mu\beta_Y}{\gamma} \quad \text{Eq. (6)"}$$

- In lines 616 to 631 we added the following paragraphs which describes the different modelling assumptions about population testing that we explored: *“We explored 4 different testing assumptions and chose the best fitting model (visually and with the lowest DIC) as the baseline model. The best fitting model assumed that only symptomatic individuals undergo diagnostic testing and that they isolate if they receive a positive result. In a second scenario, we assumed that all symptomatic individuals isolate and that asymptomatic individuals test with a probability p and isolate if they receive a positive result. In the third scenario, we assumed that symptomatic individuals test and isolate if they receive a positive result, and that asymptomatic individuals test with a probability p and isolate if they receive a positive result. In a fourth scenario we assumed that the probability of taking a diagnostic test is independent of symptom occurrence, and that asymptomatic and symptomatic individuals test with the same probability p and isolate if they receive a positive result. We also ran a thorough sensitivity analysis on the best fitting model. Firstly, we assumed that symptomatic infectious individuals who did not test or who receive a false negative test result limit their contacts by a factor $(1-\alpha)$, which was estimated from the data assuming the prior $\alpha \sim \text{beta}(1,1)$. “*

2. **Relatedly, the sensitivity of 64% for other variants is probably much higher when considering time periods when individuals are at risk of transmission.**

We thank the reviewer for highlighting the uncertainty in the sensitivity of the antigen test. We originally chose to assume an antigen test sensitivity of 64.3% based on the performance of the antigen test in the hospital surveillance study against variants that did not include the M234I-A376T mutations. However, we recognise that this figure was estimated from a limited number of samples, so we have rerun the main analysis using an antigen test sensitivity of 68.9%, which was estimated from the 1387 subjects who received both an antigen and molecular test in the hospital surveillance study. We also include sensitivity analysis assuming antigen test sensitivities of 64.3% and 87.5% against the concordant variants. We find that the posterior estimates are not significantly different to the baseline scenario, although assuming 87.5% sensitivity results in a lower estimate of the probability of a symptomatic individual testing (66.36%; 95%CrI, 25.32-99.4 vs 92.7%; 95% CrI, 72.8-99.8%). Notably, in a counterfactual analysis we find that increasing the assumed sensitivity of the antigen test from 68.9% to 87.5% increases the transmission advantage provided to the M234I-A376T variant, supporting our conclusion that antigen testing can allow for the undetected spread of a discordant variant.

Summary of changes made:

- We reran the main analysis assuming an antigen test sensitivity against the concordant variants of 68.9% and updated Figure 5 and Table 2.
- We ran sensitivity analyses assuming antigen test sensitivities of 64.3% and 87.5%, updating Supplementary Table S7 accordingly.
- In the results lines 201-3 we edited the text to reflect the updated antigen test sensitivity estimate used in the main analysis and to discuss the sensitivity analysis exploring the impact of higher and lower antigen test sensitivity estimates: *“In line with the findings from the hospital-based study described above, we assumed antigen sensitivity to be 0% against the discordant M234I-A376T variant and 68.9% against the A220V, Alpha and all other variants (i.e., the concordant variants).”*
- In the results in lines 230-238 we added the following sentences discussing the results of the sensitivity analysis: *“In a sensitivity analysis we assessed the impact of assuming lower and higher estimates of antigen sensitivity against the concordant variants; different infectious and latency periods; imperfect vaccine efficacy against infection and the extent to which symptomatic individuals limit their exposure, independently of testing. Across all scenarios and all parameters we found no significant difference in the posterior mean and 95% CrI (Supplementary Table S7). Notably, the mean probability of a symptomatic case taking a diagnostic test was 87-93% for all scenarios, with the exception of scenario 3, where we assumed antigen test sensitivity against the concordant variant to be 87.5% rather than 68.9%; under this assumption we estimated the probability of a symptomatic individual taking a diagnostic test to be lower (66.36%; 95%CrI, 25.32-99.44).”*
- In the results, lines 263 to 269 we updated the text in accordance with the updated antigen test sensitivity estimate: *“We estimate that antigen testing alone (i.e., without molecular confirmation), assuming an antigen test sensitivity of 68.9%, would result in a cumulative incidence of 40.2% (95% CrI, 36.8 - 42.4%) in Veneto. This compares with a cumulative incidence of 21.1% (95% CrI, 18.2- 25.5%) under the baseline testing scenario”*

(Figure 5d). Assuming an antigen sensitivity of 87.5% (Supplementary Tables S4 and S7) and an antigen only testing strategy increases the transmission advantage of the discordant variant further (Figure 5d)."

- In lines 379 to 381 of the discussion we include a sentence acknowledging the assumptions made about antigen test sensitivity against the concordant variants: "We also assumed a concordant test sensitivity of 68.9% based on the hospital study, although we explore the impact of higher and lower antigen test sensitivity estimates in a sensitivity analysis.."
 - In the methods lines 555-557 we updated the text as follows: "We assumed the molecular sensitivity ϕ_{MO} to be 92.0%³ against both concordant and discordant variants and the antigen test sensitivity ϕ_{AN} to be 68.9% against the concordant variants and 0% against the discordant variant (Supplementary Table S3)".
 - In lines 631 to 634 of the methods, we updated the section describing the sensitivity analyses run: "Secondly, we assessed the robustness of the results when assuming antigen test sensitivities of 64.3% and 87.5% (Supplementary Tables S3 and S4) against the concordant variant."
3. **Several of the model outputs are confusing. In particular, the underlying betas (transmission rates) for the alpha, A220V and 'other' variants is twice as high in Veneto as the rest of Italy, however the M234I-A376T transmission rate is approximately the same across the two locations. In addition, 76% of infected individuals got tested in Veneto compared to 29% in the rest of Italy. I found it difficult to come up with viable hypotheses to explain such discrepancies –that the difference in transmissibility between locations should differ by variants (especially for the key variant we are interested in), that there were such substantial differences in accessibility to tests by the end of 2020. I suspect that some of these slightly unexpected estimates may be due to the extreme modelled scenario of complete control of detected cases.**

We thank the reviewer for raising this important point, which has given us the opportunity to improve our fitting procedure and the model estimates. We now fit the models simultaneously to the data from Veneto and the rest of Italy, using a single beta parameter per variant, the same reporting rate ρ and impact of interventions ω across the two locations so that changes in the transmission dynamics are solely determined by the differences in testing strategies (proportion of antigen and molecular tests used).

Summary of changes made:

- We re-ran the analysis fitting Veneto and the rest of Italy in the same model, updating Figure 5 and Table 2 accordingly.
- In lines 221-223 of the results we updated the text to reflect that there is now only one reporting rate for both Veneto and the rest of Italy: "We estimated the probability of a symptomatic infectious individual taking a diagnostic test to be 92.7% (95% CrI, 72.8-99.8%) for Veneto and Italy."
- In lines 598-602 of the method we edited the text as follows: "We simultaneously calibrate the model to the data from Veneto and the rest of Italy by fitting to the variant-specific reported incidence relative to the population size (i.e., divided by S_0 and using a scaling factor of 100,000), assuming the same transmission

parameters β_Y , probability of a symptomatic individual taking a test ρ , and reduction in transmission ω_I across the two locations.”

- 4. The authors use different ranges when plotting the epidemic curves in Veneto and the rest of Italy, which hinders comparison in Figure 5A, but it seems that the timing of the first epidemic peak is modelled to be earlier in Veneto than the rest of the country. It would be useful to compare that with reported epidemiological data to see if that was indeed the case.**

We thank the reviewer for this useful point, the timings of the epidemic peak are indeed different in Veneto and the rest of Italy. We have now added Figure 3b which presents the observed epidemic curve in Veneto and the rest of Italy to make this clearer.

We now seed each variant in both locations one month before its detection in Italy, meaning we seed each variant at the same time across two locations and that the modelling periods are the same. We think that this makes sense as we would not expect a variant to emerge later in Veneto compared to the rest of Italy, so differences in variant detection are likely attributable to differences in genomic surveillance. This is supported by the observed data, as there are months when no genome sequences from Veneto were deposited in GISAID (July and September 2020).

Summary of changes made:

- We have added Figure 3b, which shows the reported incidence for both Veneto and the rest of Italy on the same figure to aid comparison of the timing of both peaks.
 - We added the following sentences to the methods in lines 503 to 507: *“We implemented a multi-strain model to reconstruct the transmission dynamics of the dominant variants carrying the A220V and M234I-A376T mutations, all other co-circulating variants, and the Alpha variant (B.1.1.7) for Veneto and the rest of Italy between May 2020-2021. The month prior to the detection of A220V in GISAID in Italy was chosen as the start of the modelling period.”*
 - The X axis of Figures 5a-b both now range from May 2020-21 reflecting the modelling study period and allowing for easier comparison of the timings of the outbreaks across Veneto and the rest of Italy.
- 5. There is no reference to time since symptom onset when calculating the sensitivity of the tests to the different variants. Individuals can stay PCR positive (which the authors use as the gold standard) for a long time after individual test negative on antigen tests. As the authors point out, antigen tests are a good marker of viable virus and linked to transmissibility – therefore reduced sensitivity after e.g., 10 days may not be important for transmission. The authors should set out the time since symptom onset for the variant results.**

We have reported in Supplementary S2 the symptom onset as the distance in days from the date of admission and the anonymized codes of the patients that are present in the “Antigen and Molecular” sheet of the excel S2 file.

Since the present work is a retrospective study, the obtained sequences derive from frozen samples. When available, the samples stored at -80°C have been extracted and, before proceeding with sequencing, they were amplified using our in-house system based on the CDC protocol which targets the N gene. This second testing was performed to confirm the potential degradation of the sample. The discrepancy between Ct obtained with our in-house method and the one based on the Diasorin kit can be explained by the different targets amplified between the two methods and also by the freezing-thawing process which can have affected the conservation of the samples. We have reported this on S2 excel file, results, and methods explaining the situation.

As far as symptom onset distance from the date of swab testing, discordant cases show a variable range in days, but we obtained a good viral genome coverage without gaps even for those samples that can be considered borderline (JKNGUXZXCUCG 10 days and JRXBFBTWFCKN 20 days). This suggests that the viruses were still intact and together with the low Ct values that were obtained at the time of diagnosis indicates infectious virus.

In other situations where Ct values are high and the distance in days from symptom onset to swab testing is over 10 days, we agree with the reviewer that the effect of sensitivity of the antigen test prevail over the problem of disruptive mutations and that, anyway, this should not be considered important for transmission.

Summary of changes made:

- In lines 148 to 156 of the results, we added the following sentences: *“We selected 8 discordant samples (molecular +/- antigen -) and 9 concordant samples (molecular +/- antigen +, as control), with S and ORF1 Ct values ranging at the time of the nasopharyngeal swab testing from 11.6 to 26.0 (See Supplemental S2) ⁴. The observed discrepancy between the original RT-PCR values obtained on the fresh nasopharyngeal swab samples with the Diasorin kit (see Methods) and the RT-PCR values obtained from the in-house molecular method on stored swab samples at -80°C, can be explained by the assay performance, including the different targets amplified, and the freezing-thawing process which may have affected the conservation of the samples (see Methods).”*

6. **Overall, the weaknesses in the model detract from the paper. At the very least, I would plot the observed proportion that were of each variant in 5C and not just rely on the model outputs. This would provide much more direct evidence of a difference on the epidemic trajectories from the antigen testing sensitivity.**

Thank you for this suggestion, we have now included figures of the GISAID prevalence of each variant as well as the reconstructed-variant specific incidence, obtained by multiplying the total incidence of positive cases by the prevalence of each variant in GISAID.

Summary of changes made:

- We added Figures 3g-3i which plot the observed prevalence of each variant used in the modelling study, extracted from GISAID data.
- We added Figures 3c-3f which plot the reconstructed-variant specific incidence, obtained by multiplying the total incidence by the prevalence of the variant in GISAID.
- In the results section lines 223 to 229, when discussing the results presented in Figure 5c (which presents the probability of detecting each variant by location) we include a sentence highlighting the observed rise in M234I-A376T frequency in Veneto, compared with the rest of Italy, not seen with the concordant variants: *“In Veneto, we estimate that the probability of detecting an M234I-A376T variant case was significantly lower than the probability of detecting a concordant variant (Figure 5c). For the rest of Italy, we estimate that the probability of detecting the M234I-A376T variant is comparable to the probability of detecting the concordant variants (Figure 5c). This is in agreement with the observed prevalence of the M234I-A376T variant, which was higher in Veneto compared with the rest of Italy (Figure 3g), a trend not seen with the concordant variants (Figures 3h-j).”*

7. Secondly, I would see if there was a difference in proportions of each variant between older and younger individuals in Veneto – as younger individuals all got PCR tested, you should see a reduced proportion of M234I-A376T in the younger age groups than older ones (assuming age-dependent mixing).

This is an interesting suggestion but the data on variant prevalence from GISAID does not include the age of the individual from whom the sample is taken, so we are not able to investigate this further. We have included a sentence in the discussion of discussing this limitation.

Summary of changes made:

- In the discussion lines 388-390, we add the sentence: *“Finally, we assumed homogeneous mixing within the population as the lack of age-specific surveillance data limited the extent to which we could account for heterogeneous contact patterns by age, for instance.”*

Reviewer #2 (Remarks to the Author):

Del Vecchio and colleagues analyse the emergence and spread of a SARS-CoV-2 variant escaping antigenic testing in Italy. The reconstruction of the variant spread in the Veneto region, compared to the whole Italy, joint with the analysis of counterfactual scenarios show how the massive use of antigenic testing could favour the emergence of escaping variants. This unintended consequence of antigenic testing calls for the design of testing strategies that combine in a reasoned way antigenic and molecular testing.

The study has important public health implications. The frequent emergence of Variants of Concern has shown the importance of molecular testing and full genome sequencing to monitor virus circulation. This work adds an important and original point to the discussion by unveiling the risk of a testing strategy too centred on antigenic testing.

The analysis presented in the manuscript is extensive and complete. To the best of my understanding it is sound and provides a strong support to authors' hypothesis. Still, some improvements are needed before I can recommend the acceptance of the manuscript.

We thank the reviewer for their positive feedback and for their helpful suggestions. We provide a response to each point raised below.

- 1. Overall the manuscript reads well. However, the presentation of the work is not always easy to follow, because the analysis is rich and the manuscript dense. I believe authors should make an extra effort to make the study more accessible to a wider audience.**

We thank the reviewer for highlighting the need to improve the accessibility of the manuscript. We have revised the text to improve clarity, separating the different parts of the study into subsections. We have now included a conceptual representation of the model through a simplified flow diagram of the compartmental transmission model in the main text (Figure 4a), moved the fully specified flow diagram in the SI (Figure S6) and included plots of the raw data (Figure 3) used (i.e., the proportion of COVID-19 patients receiving antigen diagnostic tests, the reported case incidence, the GISAID prevalence of each variant and the reconstructed-variant specific incidence, obtained by multiplying the reported case incidence by the prevalence of the variant in GISAID).

Summary of changes made:

- We have added Figure 3 to help the reader visualise the data used for fitting.
- In Figure 4a we present a simplified flow diagram of the model structure.
- We have worked on the text throughout the manuscript to try and improve the coherence and accessibility of the manuscript.
- We have added a separate subsection on sensitivity analysis in the methods.
- We have updated Tables 2, Supplementary Table 6 and Supplementary Table 7 to present the R_0 estimates of each variant, rather than the β estimates as they are more intuitive to interpret.

- 2. The brief synthesis presented at the end of the Introduction section could be extended to better anticipate the different parts of the analysis and guide the reader throughout the results.**

Thank you for the constructive comment, we have extended the introduction as suggested, to better explain the different sections of the study.

Summary of changes made:

- In the introduction lines 91-111 we have added: *“Here we first present the results of a hospital-based surveillance study in the Veneto region (Italy), during which we identified a viral variant that escapes detection by antigen tests, characterised by multiple disruptive amino-acid substitutions in the N antigen. As this variant was found to be circulating at a higher frequency in Veneto, where 57% of tests conducted between September 2020 and May 2021 were antigen, than the rest of Italy, where 35% of tests conducted were antigen⁵, we next test the hypothesis that the increased frequency of antigen testing in Veneto compared to the rest of Italy could have favoured the undetected transmission of the discordant variant. To this end, we fit a multi-strain compartmental model of SARS-CoV-2 transmission, which accounts for population testing, vaccination, and non-pharmaceutical interventions, to the epidemiological and genomic data recorded in Veneto and in the rest of Italy, allowing us to reconstruct variant-specific incidences. We additionally test the impact of several counterfactual testing scenarios on the transmission dynamics, diagnostic test performance and genomic surveillance of escaping and concordant SARS-CoV-2 variants. Together, our results shed new light on the limitations of mass antigen testing in the absence of molecular testing, and on the importance of maintaining molecular testing, not just for diagnostic but also for monitoring and surveillance purposes.”*

3. The different subsections of the Results section could be better linked one to the other, e.g. with sentences places at the beginning of subsections to anticipate their aim.

Thank you for the suggestion, we have added in sentences at the beginning of the results subsection to introduce the reader to the section and better explain the aims and rationale.

Summary of changes made:

- At the start of the section “Analysis of antigen assay performance”, we have edited the text as follows: *“Between 15th September 2020 to 16th October 2020, we conducted a hospital surveillance study at the University Hospital of Padua (Italy) to compare the performance of SARS-CoV-2 rapid antigen tests against RT-PCR molecular swabs. During the hospital study, 1,441 subjects representing 44% of all patients examined (3,290) in the Emergency and Infectious Diseases wards were tested with both antigen (Abbott) and molecular (Simplexa™ COVID-19 Direct Kit, Diasorin Cypress, CA, U.S.A.) tests (Figure 1).*
- At the start of the section “Reconstructing the transmission dynamics of concordant and discordant variants” we have extended the introduction to more thoroughly introduce the model: *“As the use of antigen diagnostic tests began earlier, and remained higher, in Veneto compared to the rest of Italy (Figure 3a), we next investigated the role of population testing in the transmission dynamics of a discordant variant (M234I-A376T) compared to concordant variants (A220V, the Alpha VOC and an ensemble of all other variants). We fit a multi-strain compartmental model (Figure 4a) to the reconstructed variant-specific reported incidence in*

Veneto and the rest of Italy (Figure 3c-3f), obtained by multiplying the total reported incidence in each location (Figure 3b) by the prevalence of the variant in GISAID, in each location (Figure 3g-3j). The antigen and molecular testing policy implemented in Italy during the modelling period (May 2020-21), and reconstructed within the model, is presented in Figure 4b. In line with the findings from the hospital-based study described above, we assumed antigen sensitivity to be 0% against the discordant M234I-A376T variant and 68.9% against the A220V, Alpha and all other variants (i.e., the concordant variants).”

- At the start of the section “Counterfactual analysis: impact of alternative testing strategies on cumulative incidence” we have edited the text as follows: “To investigate the impact of alternative testing strategies on the transmission dynamics of the concordant and discordant variants in Veneto, we explored several testing counterfactual scenarios (Figure 5d).”
- At the start of the section “Diagnosis of concordant and discordant variants under alternative testing strategies” we have edited the text as follows: “Given the impact of molecular and antigen testing strategies on the transmission dynamics of the concordant and discordant variants, we next investigated the performance of testing regimes in terms of PPV, NPV, and the probability of testing positive $p(T+)$, given the assumed prevalence of concordant and discordant variants (Figure 6).”
- At the start of the section “Genomic detection of concordant and discordant variants under alternative testing strategies” we have edited the text as follows: “Finally, we investigated the impact of antigen and molecular testing scenarios on the detection of variants through genomic surveillance, assuming that 0-3% of samples are selected for genomic sequencing..”

4. I also have specific remarks on the presentation of the methods:

- 1) the different parts of the analysis refer to different study periods. However, study periods are not always clearly indicated in the text.

Thank you for pointing out that this part was not clear, we have amended the text and all references to the study period to make this clear.

Summary of changes made:

- Throughout the manuscript, we now refer to either the hospital study period or the modelling study period.
- 2) Authors write in the main text “the temporal dynamics of the reproductive number of each variant are described in Supplementary Figure 7”. From this sentence I understand that the figure shows the R_t index – as can be computed with the Cori et al method. However, from a close look to the figure this does not seem the case and the quantity plotted seems, instead, the one defined in Eq. (6). Authors should be clearer on this

Apologies for the confusion. Figure S7 presents the basic reproduction number R_0 as defined in eq. 6 of the Methods in the main text, it does not represent the R_t obtained from Cori et al. The purpose of this figure is to

show how the R_0 of each variant is a function of the relative levels of antigen and molecular testing over time, however we agree that this was not clear. We have now removed the figure and instead present the R_0 estimates in Table 2.

Summary of changes made:

- We have removed Supplementary Figure 7 and updated Table 2 to present the R_0 estimates.

3) is ω_2 (the percent of reduction in transmission after control measures for the third wave) applied to the initial transmissibility or to the transmissibility already reduced by ω_1 ?

ω_2 is applied to the initial transmissibility, we agree that this was not clear from the text so have amended the methods accordingly. Essentially, we are estimating the reduction in transmission during the second and third waves independently, as the interventions implemented in the second wave were not carried over throughout the study period into the third wave.

Summary of changes made:

- In lines 576-9 of the methods section we edited the text as follows: *“we assumed that the reproduction number of all variants at the start of the start of the modelling period was reduced by a factor $(1 - \omega_1)$ from the 15th of November 2020 – 14th March ($l = 1$) and reduced by a factor $(1 - \omega_1)$ from the 15th of March 2021 onwards ($l = 2$), for both Veneto and the rest of Italy”*

5. Figures should be improved. In particular Figure 4 is not easy to read – especially panels b-e. Curves in panels a and b of Figure 5 are not clearly visible. In general, readability of figures should be improved by adding legends and panels’ titles directly to the figures when missing – e.g. description of line styles in Figure 6; legends and panels titles in Figure S4.

Thank you for highlighting the need to improve the figures. We agree that Figure 4 was not aiding the readers understanding of the model so have removed panels b-e. Instead, we have now include a simplified conceptual figure of the compartmental model and we have also added a visualisation of the epidemiological data from Veneto and the rest of Italy (Figure 3). We have also made aesthetic edits to Figures 5-7 and Figure S4 to improve their readability and have made all figures self-standing by adding legends and titles as suggested.

Summary of changes made:

- We have removed Figures 4b-4e.
- We added a new Figure 4b which details the main compartmental transmission model.

- We have added Figure 3 to better show how the variant specific incidences were reconstructed from case and genomic prevalence data.
- We have changed the colour scheme used in panels a and b of Figure 5 to allow for better visual distinction of the different curves in the plot.
- We have added figure legends to all panels of Figure 5.
- We have updated the Y axis of Figure 5c-e.
- We have added figure legends of the line type for both Figures 6 and 7.
- We have added legends and panel titles to Figure S4.

6. **Authors provide some sensitivity analysis to the multi-strain model. Still, the model relies on the choice of several parameters. I believe authors should consider expanding the sensitivity analysis (e.g. incubation and infection period, effectiveness vaccination) to show that their results are robust to different modelling assumptions.**

Thank you for the suggestion. We now include several additional sensitivity analyses to explore the robustness of the choice of parameters indicated by the reviewer, as well as other assumptions.

Firstly, as explained in greater detail in point 1 to reviewer #1 we now explore 4 initial model variants of testing, which relaxes some of the modelling assumptions made previously by accounting for pre-symptomatic, symptomatic and asymptomatic transmission. We selected scenario 1, which assumes that only symptomatic cases test and isolate, as the main model as it had the lowest DIC. We present the other model variants in Supplementary Table 6 for comparison.

In addition to the pre-existing sensitivity analysis where we assume a higher antigen test sensitivity against the concordant variants (87.5%) than the main analysis (now assumed to be 68.9%, as estimated in the hospital surveillance study), we now include a sensitivity analysis where we assume a lower antigen test sensitivity against the concordant variants (64.3%).

In the main analysis we previously assumed an incubation period of 5.1 days and an infectious period of 2.1 days. We have now extended the main model to account for pre-symptomatic transmission (as explained in greater detail in our response to point 1 of reviewer #1). As a result, the main analysis now assumes an average latency period of 1.31 days¹, a pre-symptomatic infectious period of 3.79 days (together making 5.1 days from infection to symptom onset²) and an average infectious period of 2.1 days¹. In a sensitivity analysis, we assume an average latency period of 2.15 days¹, a pre-symptomatic infectious period of 2.95 days (together making 5.1 days from infection to symptom onset²) and an average infectious period of 2.1 days¹. In a second sensitivity analysis, we assume 5.6 days from infection to symptom onset⁶ and an average latency

period of 1.31 days², an average pre-symptomatic infectious period of 4.29 days¹ and an average infectious period of 1.6 days¹.

To account for uncertainty in vaccine efficacy, we now include a sensitivity analysis where we allow for imperfect vaccine efficacy estimates against infection (70%, 80%)⁷. However, we note that by the end of the modelling study period, only ~20% of individuals had been double vaccinated, so we do not expect vaccination have a substantial impact on our model. Finally, we also included an additional analysis where we assume that symptomatic individuals who do not test, or who receive a false negative, still reduce their exposure by a factor a , which we estimate.

Notably, all sensitivity analyses find comparable posterior mean and 95% CrI. The most notable analysis assumed that antigen test sensitivity was 87.5% against concordant variants, as we estimated the probability of symptomatic individuals testing p to be 66.36% (95%CrI, 25.32-99.44) compared with the baseline scenario and other sensitivity analyses which estimated a high probability of symptomatic individuals testing (posterior means ranging from 87-93%). Notably however, when we ran counterfactual testing scenarios later in the study, we find that assuming a higher antigen test sensitivity against the concordant variants increases the transmission advantage of the escaped antigen variant further, supporting the main conclusions of our paper.

Together, these total 4 model variants with an additional 7 sensitivity analyses. We believe these additional analyses enable us to explore the key assumptions made in our model, including testing scenario, vaccination, antigen test sensitivity and the duration of the latency and infectious periods of COVID-19.

Summary of changes made:

- We ran the sensitivity analyses outlined above and present results in Supplementary Tables S6 and S7.
- In lines 205 to 209 of the results we added the following paragraph discussing the model variants: *“We fit 4 different model variants of population testing, varying assumptions on access to testing (symptomatic vs. asymptomatic cases), as well as the probability of isolating given a false negative result (See Methods for a full description). Supplementary Table S6 shows the posterior mean, 95% CrI and DIC (Deviance Information Criterion) of each model. The best fitting model assumed that only symptomatic cases test and isolate, and is thus presented as the main analysis.”*
- In lines 230-238 of the results we add the following paragraph discussing the sensitivity analysis: *“In a sensitivity analysis we assessed the impact of assuming lower and higher estimates of antigen sensitivity against the concordant variants; different infectious and latency periods; imperfect vaccine efficacy against infection and the extent to which symptomatic individuals limit their exposure, independently of testing. Across all scenarios and all parameters we found no significant different in the posterior mean and 95% CrI (Supplementary Table S7). Notably, the mean probability of a symptomatic case taking a diagnostic test was 87-93% for all scenarios, with the exception of scenario 3, where we assumed antigen test sensitivity against*

the concordant variant to be 87.5% rather than 68.9%; under this assumption we estimated the probability of a symptomatic individual taking a diagnostic test to be lower (66.36%; 95%CrI, 25.32-99.44)."

- In lines 616 to 642 of the methods, we extended the section on sensitivity analyses as follows: *"We explored 4 different testing assumptions and chose the best fitting model (visually and with the lowest DIC) as the baseline model. The best fitting model assumed that only symptomatic individuals undergo diagnostic testing and that they isolate if they receive a positive result. In a second scenario, we assumed that all symptomatic individuals isolate and that asymptomatic individuals test with a probability ρ and isolate if they receive a positive result. In the third scenario, we assumed that symptomatic individuals test and isolate if they receive a positive result, and that asymptomatic individuals test with a probability ρ and isolate if they receive a positive result. In a fourth scenario we assumed that the probability of taking a diagnostic test is independent of symptom occurrence, and that asymptomatic and symptomatic individuals test with the same probability ρ and isolate if they receive a positive result.*

We also ran a thorough sensitivity analysis on the best fitting model. Firstly, we assumed that symptomatic infectious individuals who did not test or who receive a false negative test result limit their contacts by a factor $(1-\alpha)$, which was estimated from the data assuming the prior $\alpha \sim \text{beta}(1,1)$. Secondly, we assessed the robustness of the results when assuming antigen test sensitivities of 64.3% and 87.5% (Supplementary Tables S3 and S4) against the concordant variant. Thirdly, we explored the sensitivity of the results on the assumptions made about the average latency period, pre-symptomatic infectious period and infectious period which were respectively fixed to (i) 1.31 days¹, 4.29 days⁶ and 1.6 days¹ and (ii) 2.15 days¹, 2.95 days² and 2.1 days¹. Finally, we evaluated the impact of assuming 100% vaccine efficacy ζ against infection following 2 vaccine doses by exploring the cases $\zeta = 70\%$ and 80% against infection⁷.

7. **The Discussion is well articulated. However, limitations are missing. One limitation that should be discussed is the fact that sequencing is not always aleatory and certain viruses are prioritised over others – based for instance on molecular testing characteristics, as authors note. This could bias the derivation of variant-specific incidence – although I suspect this to affect mainly the incidence of the Alpha variant.**

We thank the reviewer for their positive feedback and have now added a discussion of the limitations in the model and in the data, as suggested.

Summary of changes made:

- In the discussion, lines 374 to 390, we added the following paragraph outlining key assumptions and limitations: *"Several assumptions underpin this study. Firstly, we assumed that antigen test sensitivity is 0% against the discordant variant, in agreement with our observations and another study reporting variants with 0% antigen test sensitivity⁸. However, it is possible that some antigen tests may detect a discordant variant if they use antibodies that recognise different N antigen epitopes or target different proteins all together. In this instance, the performance of antigen testing strategies against discordant variants would be >0%. We also assumed a concordant test sensitivity of 68.9% based on the hospital study, although we explore the impact of*

higher and lower antigen test sensitivity estimates in a sensitivity analysis. We also assumed that the prevalence of a variant in GISAID is representative of its population-level prevalence, which may not always be the case. For instance, the Alpha variant may have been oversampled due to its characteristic S-gene-target-failure, further highlighting the need for robust and unbiased genomic surveillance systems. In our main analysis we assumed that only symptomatic individuals undertake diagnostic testing, however in a sensitivity analysis we explored alternative testing scenarios. Finally, we assumed homogeneous mixing within the population as the lack of age-specific surveillance data limited the extent to which we could account for heterogeneous contact patterns by age, for instance.”

Minor revisions:

1. **“Some of the supernatants were evaluated with.”: This sentence is misplaced or unclear**

Thank you for highlighting this. The incomplete sentence was a typo and has been removed.

2. **“Sequence data and corresponding metadata, released from 17th of February 2021”: Should it be 2020?**

Thank you for spotting this typo. We have completed the sentence with the full date range.

Summary of changes made:

- In lines 488-9 of the methods we have added the following text: *“released from the 1st of January 2020 to the 17th of February 2021”*

3. **“We obtained data on the mean daily number of second doses administered in Veneto and the rest of Italy ...”: No reference to vaccination was done up to this point. As such the sentence is confusing.**

Thank you for pointing this out, we have now mentioned the vaccination campaign and its timing in the introduction and methods sections.

Summary of changes made:

- In the Introduction lines 103-6 we have added the following text: *“To this end, we fit a multi-strain compartmental model of SARS-CoV-2 transmission, which accounts for population testing, vaccination, and non-pharmaceutical interventions, to the epidemiological and genomic data recorded in Veneto and in the rest of Italy, allowing us to reconstruct variant-specific incidences.”*
- In the Methods lines 534 to 537 we have edited the text as follows: *“To account for the SARS-CoV-2 vaccination campaign which began in December 2020, we obtained data on the mean daily number of second doses administered in Veneto and the rest of Italy in month m V_m from the Extraordinary Commissioner for the Covid-19 emergency⁹.”*

4. “We assumed that 98% and 97% of the population of Veneto and the rest of Italy were susceptible to SARS-CoV-2 as of July and May 2020”: Maybe July and May were inverted?

Thank you for highlighting that this sentence was not clear. The figures referred to the percentage of the population in Veneto (98%) and the rest of Italy (97%) that were susceptible at the start of the modelling periods (July for Veneto and May for Italy) in the original version of the model. However, we have now updated the model to seed all variants in both Veneto and the rest of Italy at the same time (a month prior to their detection in GISAID in Italy which is May 2020 for A220V and other variants, August 2020 for M234I-A376T and November 2020 for Alpha). We have updated the text to reflect this.

Summary of changes:

- We have edited lines 542 to 547 of the methods as follows: *“In accordance with the results reported in the national seroprevalence survey in July 2020¹⁰, we assumed that 98% of the population of Veneto and 97% of the population of the rest of Italy were susceptible to SARS-CoV-2 at the start of the modelling period (i.e., were in the S compartment, $S_0 = 4,753,625$ for Veneto and $S_0 = 53,021,564$ for Italy).”*
- 5. “The monthly prevalence trend of the variants found in concordant and discordant cases was plotted from January to December 2020 and reported for Europe, Italy, and the Veneto region (Supplementary Figure S5).”: Only Italy and the Veneto region are reported in Figure S5.**

We thank the reviewer for spotting this typo, we have removed the word Europe from the sentence.

Reviewer #3 (Remarks to the Author):

Thank you for the opportunity to review this manuscript. The reporting of novel COVID-19 variants circulating within Veneto that are not detected by current rapid tests is an important and highly relevant finding. The authors produced a mathematical model to illustrate how various testing strategies may select for these novel variants, the results of which can be used to help inform policy.

Overall, I found the introduction well written, providing the necessary context for the study. The results were detailed, but sometimes difficult to understand, without first reviewing the methods and supplementary material. The discussion was again well written, but could have included additional information (see comments below).

We thank the reviewer for their positive feedback and constructive suggestions. We have provided response to each point raised below.

Further specific comments:

1. **Methods (study design) and results (lines 103-110).** The study design states that all subjects were tested with both antigen and molecular assays. However, line 314 states that individuals from 0-19 years were predominantly tested with molecular swab, with few tested by both assays. I do not understand; if the study design was to test with both antigens, why did this not apply to a certain age group? The study design section implies that the sampling protocol was specific for this study (eg "Swab samples were collected from 15th September 2020 to 16th October 2020 from patients ..." and " All subjects were tested within one hour with both antigen..." and "Information was collected from each individual about their age, sex, date of sampling, symptoms..."). However the ethics statement reads "The data were extracted from the electronic medical record administrative database". The authors should clearly state whether the data were prospectively collected so that they had control over the test protocol, or whether the data were extracted from routinely collected clinical data.

This was a retrospective study conducted on routinely collected clinical data, in compliance with local regulations (which considered double swab testing an invasive diagnostic technique for the 0-19 years age-group) and the school admission policy (negative molecular test for school re-admission after the onset of symptoms or identification as close contact by contact tracing) in place at the time. This explains why the vast majority of the patients in age group 0-19 years was tested with a molecular test instead of a molecular and antigenic test. In this scenario and in the context of this retrospective study, we were not able to obtain sufficient data from teens admitted to hospital wards for swab testing because both antigen and molecular tests were limited to very specific clinical cases hence possibly source of biases.

We have clarified these details in the text as summarised below.

Summary of changes made:

- In the methods lines 403-404 we highlight that the study was retrospective: *"This study is a retrospective analysis of routinely collected data including swab samples collected from 15th September 2020 to 16th October 2020"*
- In lines 370-372 of the methods we change the text to states that subjects >19 years received both tests: *"All subjects > 19 years were tested within one hour with both antigen (Panbio™ COVID-19 Ag Rapid Test Device, ABBOTT Lake Country, IL, U.S.A.) and molecular (Simplexa™ COVID-19 Direct Kit, Diasorin Cypress, CA, U.S.A.) swab assay"*
- In lines 510-512 we edit the methods as follows: *"Individuals 0-19 years old were predominantly tested with molecular swabs only in compliance with local regulations and the school admittance policy (negative molecular test for re-admission to school upon a) or b), as described above) in place at the time; the few that were tested with both antigen and molecular swabs were omitted from the analysis to avoid sampling biases."*

- 2. Methods (line 323). It is not clear how the results from the in-house real-time RT-PCR informed the study. Were samples that had a single positive target on the molecular assay, but a positive RT-PCR result classified as positive? Or were only samples with a positive result for both viral targets included, irrespective the of the RT-PCR testing?**

Thank you for highlighting that this was not clear. We used the following criteria:

- 1) Samples with a single positive target on the molecular assay, but a positive RT-PCR result, were classified as positive.
- 2) Samples with both targets identified on the molecular assays but with both Ct values > 33 were checked with our in-house RT-PCR method. If positive also for our in-house method, these samples were considered positive
- 3) Samples with both targets identified on the molecular assay and with both Ct < 33 were not tested further with our in-house method and were considered positive.

Summary of changes made:

- We extended the sentence in line 428-431 of the methods to highlight how the results from the in house assay informed the study, as follows: *“Samples showing a positive result for both viral targets were considered positive. Samples with either a single positive target or with Ct value ≥ 33 were confirmed with an in-house real-time RT-PCR targeting the N2 gene¹¹, if this was also positive then the sample was considered positive.”*

- 3. The transmission model structure (Figure S6) does not include any infection of susceptibles based on the infectious reservoir. Therefore, I am unclear whether the results of the model (in terms of number of infectious individuals) informs subsequent time steps, or whether it is simply an output to compare against epidemiological data. If there is no feedback within the model then I am not sure why recovered compartment is included since the rate of recovery is assumed.**

There is feedback in the model, which is required to reconstruct the epidemic dynamics. The force of infection (λ_Y) in the transmission model structure is dependent on the proportion of Infectious (I) individuals among the initially susceptible individuals (S_0) and is defined in equation (1) in the methods as:

$$\lambda_Y = \frac{\beta_Y(IP_Y + IA_Y + IS_Y)}{S_0},$$

We have included this information in the figure caption of the flow diagram.

Summary of changes:

- In Figure 4a and Supplementary Figure 6 we edited the figure caption as follows: *“Susceptible individuals are denoted S and are infected at rate $\lambda_Y = \frac{\beta_Y(IP_Y + IA_Y + IS_Y)}{S_0}$.”*

4. Results (lines 142 & 143). The reported sensitivity values should have 95% CIs reported. Only 3 samples contained the M234I and A37T mutations, which is insufficient to make any definitive statement about test performance against this variant. The results presented also seem open to the effects of confounding and bias. Figure S3 clearly shows the test sensitivity decreases with increasing Ct, and the authors have found that discordant samples have significantly higher Ct values. In addition, the authors have elected to sample 8 discordant and 9 concordant samples, dramatically changing the sample profile where discordant samples only made up 1% of all samples. For all of these reasons it is likely the reported sensitivity of 64.3% against variants not carrying the substitution is under-estimated. The value of 64.3% sensitivity is used in the modelling for concordant variants. I am wondering why the authors have elected to use this value, which is based on only a relatively small number of tests, rather than the more robust 68.9% obtained for all samples in the hospital study.

We agree with the reviewer. Indeed, we reported only few sequence data from samples with a high sequence coverage especially in the N gene region objective of this study, to be sure not to have uncertainties in the detection of mutations. Some samples from positive patients were not of enough quality, hence discarded in this study. Moreover, we only selected samples for sequencing with low ORF1 and S gene Ct values, indicating that the samples were discordant owing to the disruptive amino acid substitution, rather than due to low viral loads.

We are aware of the limited sampling for sequencing. Despite this, we were able to identify the unexpected recurrence of the double mutation M234I-A37T. This haplotype was found to have a higher incidence in Veneto compared to the rest of Italy, suggesting its escaping property as opposed to higher infectivity. Nonetheless, we agree that the sensitivity figure used in the modeling work should be robust and we have changed it accordingly to use the estimate of 68.9%. We also include a sensitivity analysis assuming 87.5% and 64.3% antigen test sensitivities against the concordant variants. We find that the posterior estimates are not comparable to the baseline scenario, although assuming 87.5% sensitivity results in a lower estimate of the probability of a symptomatic individual testing (66.36%; 95%CrI, 25.32-99.4 vs 92.7%; 95% CrI, 72.8-99.8%). Notably, we find that assuming a high antigen test sensitivity against the concordant variant increase the final size of the epidemic owing to the additional transmission advantage provide to the discordant variant, in support of the main conclusions of the paper.

We have also now included sentences in the discussion acknowledging the limitations of the antigen test sensitivity estimates used in the study and discussing the potential impact of these assumptions.

Summary of changes made:

- We reran the main analysis assuming an antigen test sensitivity against the concordant variants of 68.9% and updated Figure 5 and Table 2 accordingly.

- In the results line 168-172 we edited the text to include 95% confidence intervals for the estimates of antigen test sensitivity against the concordant variants: *“From the samples sequenced, we estimate the Panbio™ COVID-19 antigen test sensitivity to be 0% (95% CI, 0-56.1%) against viruses with the M234I-A376T substitutions and 64.3% (95% CI, 38.8- 83.7%) against variants not carrying the M234I-A376T substitutions (Supplementary Table S3). When considering only the viral variants without mutations in the regions of mapped B cell epitopes, we find a test sensitivity of 87.5% (95% CI, 52.9-99.4%) (Supplementary Table S4).”*
- In the results lines 201-203 we edited the text to reflect the updated antigen test sensitivity estimate used in the main analysis and to discuss the sensitivity analysis exploring the impact of higher and lower antigen test sensitivity estimates: *“In line with the findings from the hospital-based study described above, we assumed antigen sensitivity to be 0% against the discordant M234I-A376T variant and 68.9% against the A220V, Alpha and all other variants (i.e., the concordant variants).”*
- In the results lines 230-238 we include a paragraph on the sensitivity analysis: *“In a sensitivity analysis we assessed the impact of assuming lower and higher estimates of antigen sensitivity against the concordant variants; different infectious and latency periods; imperfect vaccine efficacy against infection and the extent to which symptomatic individuals limit their exposure, independently of testing. Across all scenarios and all parameters we found no significant difference in the posterior mean and 95% CrI (Supplementary Table S7). Notably, the mean probability of a symptomatic case taking a diagnostic test was 87-93% for all scenarios, with the exception of scenario 3, where we assumed antigen test sensitivity against the concordant variant to be 87.5% rather than 68.9%; under this assumption we estimated the probability of a symptomatic individual taking a diagnostic test to be lower (66.36%; 95%CrI, 25.32-99.44).”*
- In the results, lines 263 to 269 we updated the text in accordance with the updated antigen test sensitivity estimate: *“We estimate that antigen testing alone (i.e., without molecular confirmation), assuming an antigen test sensitivity of 68.9%, would result in a cumulative incidence of 40.2% (95% CrI, 36.8 - 42.4%) in Veneto. This compares with a cumulative incidence of 21.1% (95% CrI, 18.2- 25.5%) under the baseline testing scenario (Figure 5d). Assuming an antigen sensitivity of 87.5% (Supplementary Tables S4 and S7) and an antigen only testing strategy increases the transmission advantage of the discordant variant further (Figure 5d).”*
- In the discussion lines 374 to 381 we added the following sentence discussing the potential impact of assumptions made in the study regarding antigen test sensitivity : *“Firstly, we assumed that antigen test sensitivity is 0% against the discordant variant, in agreement with our observations and another study reporting variants with 0% antigen test sensitivity 23. However, it is possible that some antigen tests may detect a discordant variant if they use antibodies that recognise different N antigen epitopes or target different proteins all together. In this instance, the performance of antigen testing strategies against discordant variants would be >0%. We also assumed a concordant test sensitivity of 68.9% based on the hospital study, although we explore the impact of higher and lower antigen test sensitivity estimates in a sensitivity analysis.”*
- In the methods, lines 555-557 we updated the text as follows: *“We assumed the molecular sensitivity ϕ_{MO} to be 92.0%³ against both concordant and discordant variants and the antigen test sensitivity ϕ_{AN} to be 68.9% against the concordant variants and 0% against the discordant variant (Supplementary Table S3).”*

5. **There seems to be a conflict in the reporting of the statistical test used to compare the Ct values between discordant and concordant samples. The methods and Table S2 state Wilcoxon-Mann-Whitney two-sample test was used, but the results report a Welch two-sample t-test. If the Mann-Whitney was used then it would be more appropriate to report the median values in Table S2, rather than the mean.**

Apologies for the confusion, and thank you for bringing this our attention. We confirm that we conducted the Welch two-sample t-test throughout and amended the text accordingly

6. **Methods (line 429). The proportion of undetected infectious individuals on any day is calculated as the number in this category divided by S₀. It is unclear to this reviewer why S₀ is used as the denominator, and not S_i since the susceptible pool would be expected to decrease over time.**

We thank the reviewer for highlighting that equation 4 is not clear. $p_{c,i}$ is the probability that an individual is infected by a concordant variant on day i , obtained by dividing the total concordant incidence on day i by the population size S_0 . We calculate this in order derive the number of positive antigen tests on a given day.

Summary of changes:

- In line 569 of the methods, we edited the text as follows: *“Let $p_{c,i}$ denote the per capita probability of being infected with a concordant variant.”*

7. **Discussion. The discussion is well written but lacks any mention of model assumptions or limitations and how these may impact the results presented.**

Thank you for the positive feedback, we agree that discussing model assumptions and limitations is useful. We have extended the discussion to highlight key assumptions and limitations of the model, including those regarding antigen test sensitivity outlined above.

Summary of changes made:

- We added the following paragraph discussing key assumptions and limitations of the model to the discussion, lines 374 to 390: *“Several assumptions underpin this study. Firstly, we assumed that antigen test sensitivity is 0% against the discordant variant, in agreement with our observations and another study reporting variants with 0% antigen test sensitivity⁸. However, it is possible that some antigen tests may detect a discordant variant if they use antibodies that recognise different N antigen epitopes or target different proteins all together. In this instance, the performance of antigen testing strategies against discordant variants would be >0%. We also assumed a concordant test sensitivity of 68.9% based on the hospital study, although we explore the impact of higher and lower antigen test sensitivity estimates in a sensitivity analysis. We also assumed that the prevalence of a variant in GISAID is representative of its population-level prevalence, which may not always be the case. For instance, the Alpha variant may have been oversampled due to its characteristic S-gene-*

target-failure, further highlighting the need for robust and unbiased genomic surveillance systems. In our main analysis we assumed that only symptomatic individuals undertake diagnostic testing, however in a sensitivity analysis we explored alternative testing scenarios. Finally, we assumed homogeneous mixing within the population as the lack of age-specific surveillance data limited the extent to which we could account for heterogeneous contact patterns by age, for instance.”

8. For instance, how realistic is it that detected infected individuals do not contribute to any transmission; they may go into isolation after detection, but this does not mean they are not infectious prior to detection.

We thank the reviewer for raising this point. To account for this observation, we have now extended the compartmental transmission model to include a pre-symptomatic compartment, where individuals are infectious. As discussed in greater detail of point 1 to reviewer # 1, we now assume that individuals progress from the exposed compartment to the pre-symptomatic compartment at a rate of 1/1.31 days, parameterised from *Lavezzo et al.*¹. We further divide the infectious compartment into symptomatic and asymptomatic, assuming that 59% of individuals are symptomatic, again parameterised from *Lavezzo et al.*¹. In our main model we assume that only symptomatic individuals test and isolate if they receive a positive test result. Therefore, our main model now assumes that both pre-symptomatic and asymptomatic individuals contribute to the transmission of SARS-CoV-2.

Summary of changes made:

- We extended the transmission model to include a pre-symptomatic, asymptomatic and symptomatic infectious compartments, rather than a single infectious compartment.
- We re-ran the analysis and updated Figure 5 and Table 2 accordingly.
- In the methods in lines 539 to 541 we edited the description of the model compartmental as follows: *“Supplementary Figure S6 shows the flow diagram of the multi-strain susceptible (S), exposed (E), pre-symptomatic infectious (IP), asymptomatic infections (IA), symptomatic infectious (IS), quarantined (Q), recovered (R) compartmental model.”*
- In the methods in lines 547 to 549 we updated the following sentence describing the transmission model: *“We assumed an average latency period 1/η of 1 / 1.31 days², pre-symptomatic infectious period 1/σ of 1 / 3.79 days¹ and an average infectious period 1/γ of 2.1 days¹.”*
- In lines 551 to 554 of the methods we edited equation 1, to reflect the contribution of pre-symptomatic individuals to the transmission of SARS-CoV-2: *We defined a variant-specific force of infection (i.e., the daily per-capita risk that a susceptible individual becomes infected) λ_{Yi}, given by*

$$\lambda_{Yi} = \frac{\beta_Y (IP_{Yi} + IA_{Yi} + IS_{Yi})}{S_0} \quad \text{Eq. (1)}$$

Thus, λ_{Yi} is proportional to the transmission rate of variant Y (β_Y) and to the proportion of undetected infectious individuals of variant Y on day i ((IP_{Yi} + IA_{Yi} + IS_{Yi})/S₀). ”

- We updated equation 4 to state that the true incidence of infection is now the rate of individuals entering the pre-symptomatic infectious compartment: “

$$p_{c,i} = \frac{\eta E_{Ai} + \eta E_{Oi} + \eta E_{Ali}}{S_0} \quad \text{Eq. (4)}$$

- We updated equation 6 which defines the reproduction number:

$$R_{0Y}^0 = \frac{\beta_Y}{\sigma} + \frac{(1-\mu)\beta_Y}{\gamma} + \frac{(1-\delta_Y)\mu\beta_Y}{\gamma} \quad \text{Eq. (6)''}$$

9. It seems that double-vaccinated individuals are not able to become infected, which is clearly not the case. Would changing these assumptions change the conclusions from the model?

We thank the review for raising this point. The vaccination campaign in Italy began at the start of 2021 and only 20% of the population were double vaccinated as of May 2021, when the modelling study period ends. Moreover, only 2.5% of the population were vaccinated as of the end February 2021, so assuming that vaccination induced immunity lasts 90 days, we would not expect waning immunity to play a large role in the transmission dynamics at this stage. However, we think it would be helpful to explore whether these assumptions do have an impact on the results. Therefore, we include sensitivity analyses where we assume that vaccine efficacy against infection is 70% or 80%⁷.

Summary of changes made:

- We ran the sensitivity analyses assuming vaccine efficacy against infection to be 70% and 80%, and present the posterior mean and 95% CrI in Supplementary Tables S7.
- In lines 230 to 234 of the results we added the following sentence: *“In a sensitivity analysis we assessed the impact of assuming lower and higher estimates of antigen sensitivity against the concordant variants; different infectious and latency periods; imperfect vaccine efficacy against infection and the extent to which symptomatic individuals limit their exposure, independently of testing. Across all scenarios and all parameters we found no significant difference in the posterior mean and 95% CrI (Supplementary Table S7).”*
- In lines 640 to 642 of the methods, we included the following sentence: *“Finally, we evaluated the impact of assuming 100% vaccine efficacy ζ against infection following 2 vaccine doses by exploring the cases $\zeta = 70\%$ and 80% against infection⁷.”*

10. Another point of interest is that the fitted values for Veneto for many parameters are higher than for Italy, so it would be useful to understand whether there is any evidence to support this. For example, is there any evidence that the reduction in transmission after additional control measures was greater in Veneto than Italy overall?

We thank the reviewer for this observation. We have rerun the analysis simultaneously fitting to the data reported in Veneto and the rest of Italy. This allowed us to estimate the same variant-specific transmission rate (β), probability of testing ρ and impact of interventions on transmission ω for both settings. These changes support our key findings, as differences in the detection of the discordant and concordant variants between Veneto and the rest of Italy are now attributable to difference in antigen and molecular testing.

Summary of changes made:

- We re-ran the analysis fitting Veneto and the rest of Italy in the same model, updating Figure 5, Table 2 and Supplementary Figure 7 accordingly.
- In lines 221-223 of the results we updated the text to reflect that there is now only one reporting rate for both Veneto and the rest of Italy: *“We estimated the probability of a symptomatic infectious individual taking a diagnostic test to be 92.7% (95% CrI, 72.8-99.8%) for Veneto and Italy.”*
- In the methods lines 598-602 we added the following sentence: *“We simultaneously calibrate the model to the data from Veneto and the rest of Italy by fitting to the variant-specific reported incidence relative to the population size (i.e., divided by S_0 and using a scaling factor of 100,000), assuming the same transmission parameters β_V , probability of a symptomatic individual taking a test ρ , and reduction in transmission ω_I across the two locations.”*

Minor comments

11. Consistency in the use of terms. It would be helpful if the antigen and molecular tests were referred to in the same way through the manuscript. For instance, in the paragraph starting line 111 the following terms are used: antigen test, PanbioTM ANCOV test, ANCOV+, Abbott test.

We completely agree with the reviewer that the use of multiple terms was confusing. We have now amended the manuscript to refer to only “antigen” or “molecular” tests to improve readability and accessibility.

Summary of changes made:

- Throughout the manuscript we have changed ANCOV to read antigen and DNCOV to read molecular.

12. Line 88: “... that can escape detection by /b> antigen testing.”

Thank you for spotting this typo, we have corrected the text.

13. Figure 1. Wording of title does not seem to accurately reflect the content of the figure. Suggested amending to “Proportion of COVID-19 patients receiving antigenic diagnostic tests in Veneto...”, or similar.

Thank you for highlighting this. We have amended both the Y axis and the description in the figure legend.

Summary of changes made:

- We have amended the Y axis label of Figure 3a to read as follows: *“Proportion of COVID-19 patients receiving antigenic diagnostic tests”*.
- We have amended the figure legend of Figure 3a to read as follows: *“Proportion of COVID-19 patients receiving antigen diagnostic tests”*.

14. Figure 2. Final sentence “The under 20 subjects have not been considered in the study.” is not grammatically correct. It is not immediately clear what information is added by separating the ‘Emergency’ and ‘Infectious Disease’ subjects in one part of the flow chart. This could be removed for improved clarity. If this element stays in the flow chart then I would suggest making sure the labelling of subjects aligns with the figure title (eg Emergency vs A&E).

We agree the final sentence of the Figure 2 legend was not grammatically correct and have changed it. We reference the Emergency and Infectious Disease wards in Figure 2 as this how the study is described in both the methods and results. Nevertheless, we agree with the reviewer that the inconsistent use of Emergency and A&E is confusing, so we have edited the manuscript to only use Emergency ward throughout.

Summary of changes made:

- We have edited the figure caption of Figure 2, now Figure 1, as follows: *“Figure 1 | diagram of admissions to hospital wards: Diagram showing the total number of patients from Emergency and the Infectious Diseases wards tested with either a molecular or antigen diagnostic test and with both molecular and antigen tests in the period from the 15th of September to the 16th of October 2020.”*
- In lines 116-119 of the results we changed A&E to Emergency, so the text now reads as follows: *“During the hospital study, 1,441 subjects representing 44% of all patients examined (3,290) in the Emergency and Infectious Diseases wards were tested with both antigen (Abbott) and molecular (Simplexa™ COVID-19 Direct Kit, Diasorin Cypress, CA, U.S.A.) tests (Figure 1).”*
- In lines 403-406 of the methods, we changed A&E to Emergency, so the text now reads as follows: *“This study is a retrospective analysis of routinely collected data including swab samples collected from 15th September 2020 to 16th October 2020 from patients admitted to the Emergency and Infectious disease wards of the University Hospital of Padua (Italy) and from subjects who required SARS-CoV-2 testing”*

15. Table 1. It is not clear what “P” refers to in the second row of the table. Please include abbreviation in title or footnote. I am assuming the authors are using P to represent the prevalence of disease, since both PPV and NPV depend on the prevalence. However, I do not see the relevance of varying estimates of disease in this Table; the proportion of PCR-positive patients will provide the prevalence of disease in this cohort, which dictates the test performance. From Figure 1 I have been able to calculate the prevalence of disease in this cohort as 0.044, which aligns with the first reported value of P in Table 1. But this information should be more obvious in the manuscript. Also, the first PPV and NPV values for all data are not displaying properly (first number missing).

Apologies for the confusion. "P" means prevalence of infection and we have now substituted "P" with "Prev" in Table 1 to avoid confusion. The first reported value indeed is the prevalence in the analysed sample as taken from the hospital's administrative databases. The other two values consider the much lower prevalence's of 0.5% and 0.1%, which, at least in some periods, may be more realistic population measures than the value of 4% measured in the convenience sample of the study. These two values are meant to show that the PPV stays acceptable for a prevalence of 0.5%, but breaks down (amounting to tossing a coin) if the prevalence further decreases to 0.1%. We have inserted this information in the main text to better clarify this important aspect.

Summary of changes made:

- In lines 129-134 of the methods we added the text: *"Table 1 also presents the Positive Predictive Value (PPV) and Negative Predictive Value (NPV) for the antigen test at different prevalence values. Notably, we find that the PPV of the antigen test is acceptable given a prevalence of 4% (i.e., the prevalence in the analysed samples taken from the hospital's administrative databases), as well as at 0.5% prevalence, however given a prevalence of 0.1% the PPV is <50%."*
- We added the following text to the figure legend of Table 1: *"The PPV and NPV are presented assuming a prevalence of 0.04, as observed in in the hospital surveillance study, as well as at the lower prevalence values of 0.005 and 0.001."*
- We added changed P to Prev in Table 1 and added a row title the row title *"Prevalence"*.

16. Figure S5. G204R and R203K are listed as variants in concordant samples. While this is true, according to Figure 3 they were also seen in discordant samples. Suggest the authors include this information in the S5 figure title.

Thank you for this helpful suggestion, we have amended the figure legend accordingly.

Summary of changes:

- We have added to the figure legend of S5 the following: *"To note that A220V, G204R and R203K mutations appear in concordant and discordant samples."*

17. Figure 4. Please define GISAID.

Thanks for raising this, we have added in a description of GISAID into the legend of Figure 4.

Summary of changes made:

- We have added the following text into the legend of Figure 4: *"GISAID: an open access database of SARS-CoV-2 genomic data"*.

18. Figure 5c. From the figure title and the information within the plot I am not certain of what this sub-figure is actually showing. “Percentage of A220V, M234I-A376T, alpha, other virus variants and total cumulative incidence reported in Veneto and the rest of Italy.” This sounds like actual data, but the following sentence and data in plot indicate this is modelled output. The vertical axis is “Percent of cumulative incidence reported”. At what time point is this cumulative percent? The figure legend has no sub-plot d) but includes sub-plot f), while figure only has a-e.

We thank the reviewer for highlighting that Figure 5C is not clear. Figure 5C presents the mean probability of detecting a case during the modelling study period (obtained by dividing the estimated variant-specific reported incidence at each time point by the estimated true incidence of each variant). We have amended the Y axis, figure legend and figure description to better explain the figure.

Thank you also for noticing the typo in the letters used in the figure legend, we have now corrected this.

Summary of changes made:

- The Y axis of figure 5C is now as follows: “*Mean probability of case detection during the modelling study period*”.
- The figure legend of Figure 5C now states that the data presented are: “*A220V model, Alpha model, M234I-A376T model and Other model*”
- The figure legend now reads as follows: “*Probability of detecting cases attributable to the A220V, M234I-A376T, alpha or other virus variants in Veneto and the rest of Italy during the modelling study period.*”

19. Table S1. Please include the number of samples used for each Ct threshold.

We have inserted the requested information in Table S1.

20. Some references seem incomplete and may need a more information like a url (eg reference 24).

We thank the reviewer for spotting the incomplete references, we have now updated all references and ensured URLs are present where required.

21. Line 333. The incomplete sentence should be removed.

Yes, apologies for this typo. The incomplete sentence has been removed.

Reviewer #4 (Remarks to the Author):

This study reports detection of SARS-CoV-2 variants that escape rapid antigen tests in Veneto Italy, potentially via mutations in their N protein, highlighting the need for retaining PCR testing for virus surveillance. The manuscript is

well written, and was a pleasure to read. The study appears apt and timely, especially considering its potential usefulness to scientists, public healthcare professionals and policy makers.

We thank the reviewer for the positive feedback and useful comments and suggestions provided below.

I have only one major comment and a few minor ones.

Major comment:

- **Figure 3 is not clear. The mutations shaded in yellow and highlighted in red are present in both the ANCOV+/DNCOV+ and ANCOV-/DNCOV+ samples. Is this meant to show that they are present in both groups? If so, why is it that some of the other samples have these same mutations but they are not highlighted? For example all the samples appear to have the R203K and G204R mutations, but its only highlighted for the 1st (D1_7_B.1.1.119) and the 9th (C1_2_B.1.1.1) samples. The legend also says that "The last six sequences do not contain any mutation on N gene according to the Wuhan reference sequence", but they obviously do from the alignment shown. Also from the alignment shown, there doesn't seem to be mutations common to the ANCOV-/DNCOV group that are not also present in the ANCOV+/DNCOV+ group, and some of the sequences shown look identical (for example the 7th sample (D2_2_B.1.177) looks identical to the last sample (C_Vo_46_BB)). All of the samples, except one, contain the R209I mutation, etc. The text on page 5 says "only 2/9 of the viruses present in the concordant samples had mutations within the region of mapped B cell epitopes, and none showed amino-acid substitutions downstream to position 220", which is obviously not the case from the figure. Are there typos in the alignment, or am I missing something?**

We thank the reviewer for this point, and apologise for presenting the sequences erroneously in Figure 3. In the original version of the paper we had incorrectly entered in the text an initial draft version of the figure in place of the final version. We have now included the correct one.

Minor comments:

1. **Having figure 1 referred to in the introduction is a bit unconventional, and I wonder if it will be better to move it to the results section instead, or perhaps to a supplementary figure**

We agree with the reviewer that it is unconventional to refer to a figure in the introduction and have now removed that reference.

Summary of changes made:

- Figure 1 is now Figure 3 and is only referred to in the results section.

2. **It may be useful to state the RNA extraction method(s) used.**

We agree that this is a good idea and have edited the text accordingly.

Summary of changes made:

- We have added a sentence about RNA extraction protocol used in lines 446 to 449 of the methods as follows: *“Total nucleic acids were purified from 200 µl of nasopharyngeal swab samples and eluted in a final volume of 100 µl by using a MagNA Pure 96 System (Roche Applied Sciences). Negative extraction control was also included. Then, the extracts were treated with DNase using the DNA-free™ Kit (Ambion, Life Technologies) following the manufacturer’s instructions prior to cDNA synthesis”.*

3. From the manuscript, there is no indication that the authors included negative controls in the RNA extraction, library preparation or sequencing steps, and their criteria for handling detection of mapped reads in the negative controls.

In our laboratories, that are involved in the SARS-CoV-2 surveillance, we routinely check potential contaminations. This is required by internal hospital wards procedures for diagnostic purposes. As far as the library construction, we always perform a couple of blank samples to verify whether there are potential contaminations across samples or in the lab material. We send these samples to our external sequencing service (Polo GGB facility - Siena, Italy) as a blind test to see if we obtain reads belonging to SARS-CoV-2 suggesting either a potential contamination during library construction or during sequencing run. When obtaining the reads, we check and filter out contaminant reads vs. human genome, viral sequences, and bacteria genomes. Finally, to confirm mutations in the assembled genome, we manually investigate using a genome browser to search for a minimum sequence coverage both for forward and reverse strand reads (usually at least 5 sequences for both strands). We have added sentences in the Materials and Methods “Synthesis of cDNA and library preparation protocol” and in “Quality check and mapping of the reads”, to better explain this.

Summary of changes made:

- We edited the text in lines 451 to 452 as follows: *“ Negative controls were included also in the following steps up to sample sequencing checking for potential contaminations.”*
- We added the following sentence to the methods in lines 477 to 479: *“High-quality reads were further filtered for the presence of non-specific captured sequences from bacterial, viral, and human genomes during library construction.”*

4. How did the authors handle patients that were sampled more than once for the same test type (ANCOV or DNCOV)? I think this is not clear from the manuscript. For example, were there patients that had more than one ANCOV tests and then had DNCOV tests during the study period? Were these concordant/discordant? Etc

Being a retrospective study based on collected administrative data, we had no follow-ups. In the hospital database there were very few reports from a small number of patients who were administered more than one test either the same day (passing from one hospital ward to another) or after one week. In other cases, patients were administered swab testing (antigenic and molecular) not the same day and we discarded these

situations. We kept only data from subjects who were administered both testing (antigenic and molecular) within an hour of each other and in the same hospital ward (contemporary double swab testing for antigenic and molecular). We have added a sentence into the methods, to clarify this.

Summary of changes made:

- We edited the text in lines 412 to 413 as follows: *"No data were collected on patients receiving the same test more than once."*

5. **From the Accession numbers provided in the supplementary data, it looks like the authors have deposited the sequences to the GISAID database. It may be useful to explicitly state that this has been done in the manuscript, including the type of data deposited (consensus sequences, bam files, FASTQ files) etc.**

We have added the submission information in the "Quality check and mapping of the reads" of Materials and Methods. In GISAID we have put minimal metadata for privacy reason and following the local ethical committee's requests.

Summary of changes made:

- We added the following sentence to the methods, lines 485 to 486: *"Sequences were submitted to GISAID (GISAID accession numbers are available of supplementary material S2)."*

6. **Typo on line 119: "as shown in Supplementary Figure S4" should read "is shown in Supplementary Figure S4"** Thank you for spotting this, we have amended it.

7. **lines 186-187: "implemented in across Italy" should read "implemented across Italy"**

Thank you for spotting this, we have amended it.

8. **line 333 contains an incomplete statement: "Some of the supernatants were evaluated with."**

The incomplete sentence is a typo and has been removed.

9. **I imagine the 'TM' in "DNA-freeTM" on line 337 is meant to be a superscript**

Thank you for spotting this, we have emended the superscript TM.

10. **line 371, statement unclear: "Sequence data and corresponding metadata, released from 17th of February 2021, were downloaded from GISAID databank". Are these samples deposited to GISAID by other labs. In that case 17th February to when (or are these the samples that were available on the 17th February)?**

We agree with the reviewer. The sentence is unclear and we have amended it to present the full dates for which we downloaded data. These are sequences produced in other labs and publicly available in GISAID.

Summary of changes made:

- We have amended the methods, lines 488 to 489, as follows: “Sequence data and corresponding metadata, released from the 1st of January 2020 to the 17th of February 2021, were downloaded from GISAID databank”.

References

1. Lavezzo E, Franchin E, Ciavarella C, et al. Suppression of a SARS-CoV-2 outbreak in the Italian municipality of Vo'. *Nature* 2020; **584**(7821): 425-9.
2. McAloon C, Collins Á, Hunt K, et al. Incubation period of COVID-19: a rapid systematic review and meta-analysis of observational research. *BMJ Open* 2020; **10**(8): e039652.
3. Ibrahimi N, Delaunay-Moisan A, Hill C, et al. Screening for SARS-CoV-2 by RT-PCR: saliva or nasopharyngeal swab? Rapid review and meta-analysis. *PLOS ONE* 2021; **16**(6): e0253007.
4. World Health Organization. COVID-19 Ag Rapid Test Device (Nasopharyngeal) IFU for Panbio - EUL 0564-032-002020. <https://www.who.int/publications/m/item/EUL-0564-032-00> (accessed).
5. Presidenza del Consiglio dei Ministri - Dipartimento della Protezione Civile. Dati COVID-19 Italia. 2021. <https://github.com/pcm-dpc/COVID-19> (accessed 25th June 2022).
6. Quesada JA, Lopez-Pineda A, Gil-Guillen VF, Arriero-Marin JM, Gutierrez F, Carratala-Munuera C. Incubation period of COVID-19: A systematic review and meta-analysis. *Rev Clin Esp (Barc)* 2021; **221**(2): 109-17.
7. Ssentongo P, Ssentongo AE, Voleti N, et al. SARS-CoV-2 vaccine effectiveness against infection, symptomatic and severe COVID-19: a systematic review and meta-analysis. *BMC Infect Dis* 2022; **22**(1): 439.
8. Bourassa L, Perchetti GA, Phung Q, et al. A SARS-CoV-2 nucleocapsid variant that affects antigen test performance. *J Clin Virol* 2021; **141**: 104900.
9. Commissario straordinario per l'emergenza Covid-19 - Presidenza del Consiglio dei Ministri. Open Data su consegna e somministrazione dei vaccini anti COVID-19 in Italia. 16th November 2021 2021. <https://github.com/italia/covid19-opendata-vaccini> (accessed 3 March 2022).
10. Istituto Nazionale di Statistica. Primi risultati dell'indagine di seroprevalenza sul SARS-CoV-2, 2020.
11. CDC Centres for Disease Control and Prevention. Research use only 2019-Novel Coronavirus (2019-NCoV) Real-Time RT-PCR primers and probes2020. <https://www.cdc.gov/Coronavirus/2019-Ncov/Lab/Rt-Pcr-Panel-Primer-Probes.Html> (accessed).

REVIEWERS' COMMENTS

Reviewer #1 (Remarks to the Author):

The paper is much improved. The authors have done a very comprehensive and robust job of responding to my concerns and that of the other reviewers. I have no additional comments.

Reviewer #3 (Remarks to the Author):

The authors have done well at addressing the various comments about the initial manuscript. The modifications have increased the readability and clarified my previous points of concern. I have no further comments to add.